# For Better or for Worse, Transformers Seek Patterns for Memorization

**Madhur Panwar**[1]    **Gail Weiss**[1]    **Navin Goyal**[2]    **Antoine Bosselut**[1]

[1] EPFL        [2] Microsoft Research India

madhur.panwar@epfl.ch

## Abstract

Memorization in language models is a critical yet poorly understood phenomenon. In this work, we investigate memorization in transformer-based language models by analyzing their memorization dynamics during training over multiple epochs. We find that memorization is neither a constant accumulation of sequences nor simply dictated by the recency of exposure to these sequences. Instead, much like generalization, memorization appears to be driven by pattern recognition. Tracking memorization dynamics in mixed datasets, we observe that models memorize different sub-datasets in distinct bursts, suggesting that each subset is associated with unique underlying patterns, and that the model prefers to learn these patterns in a consistent order. We also find that easily learnable patterns tend to support generalization on unseen data, while more complex patterns do not. Furthermore, in datasets with weak or absent patterns, larger models may delay memorization relative to smaller ones, a behavior we term *overthinking*. Our results show that the subset of sequences memorized by a model over time is not arbitrary, and give insights into the internal processes a model goes through during training. Our code is available at: https://github.com/mdrpanwar/memorization-patterns.

## 1   Introduction

Memorization, particularly in the sense of outputting entire training sequences verbatim, has been observed in many large transformer models and is increasingly recognized as both a critical capability and a potential liability [Hartmann et al., 2023]. In particular, prior work has shown that certain training examples—-including those containing personal identifiers or unique texts—can be memorized and later extracted from a model's output [Carlini et al., 2023b], and the phenomenon is seen as a serious privacy concern [Yang et al., 2024].

This risk has motivated efforts to characterize and mitigate memorization in language models. For example, Carlini et al. [2023b] demonstrated that large language models often memorize parts of their training data verbatim, and that such memorization tends to grow with model size and training data duplication, meaning some sequences are memorized over others. Other studies and defenses consider memorization in a fully-trained model, defining it in various ways [Zhang et al., 2023, Carlini et al., 2023c, Nasr et al., 2023b]. However, the mechanisms of the phenomenon itself, and in particular how it emerges during training, remain underexplored.

So how does memorization happen? The taxonomy of memorization in Prashanth et al. [2025] suggests that memorization can come from either *recitation* (memorizing a sequence with no regard for its structural similarity to other data), *reconstruction* (using general patterns to fill in the gaps), or *recollection* (memorization of rare sequences). In this work, we investigate both this distinction and other mechanisms underlying memorization by tracking the memorization dynamics of different models during training.

39th Conference on Neural Information Processing Systems (NeurIPS 2025).

By tracking when and what a model memorizes during training, we gain insight into its internal learning process. We conduct experiments on data from both a real-world corpus (WikiText) and a set of synthetic datasets where we control the presence and "difficulty" of patterns in the data. The mixed dataset setting allows us to observe how a model handles easy-to-generalize data (reconstruction) versus data that is more likely to be memorized as is (recitation). We find that memorization often occurs in distinct bursts, where each burst tends to correspond to a subset of data with its own characteristic pattern. This suggests that the model is effectively segmenting the training data by patterns and fitting those subsets one at a time (reminiscent of a curriculum).

Overall, we find that memorization is far from being an arbitrary accumulation of sequences and is fundamentally tied to how models learn. Commonly, memorization in a language model is seen as the opposite extreme to that of *generalization*: the ability to extract and utilize patterns from the training data to handle unseen examples. Yet, as we find in this work, memorization and generalization may utilize the same pattern-seeking mechanisms. Transformers are highly adept at finding statistical and linguistic regularities ("patterns") in data. If meaningful patterns exist, the model will leverage them to minimize loss (this yields generalization). In fact, deep networks are known to prioritize learning simple patterns first, and only later fit noise or idiosyncratic data [Arpit et al., 2017a].

Ultimately, by understanding memorization from a training-dynamics perspective, we aim to guide future work in balancing memorization and generalization for safer and more effective language models. In particular, we suggest designing interventions targeting how models detect patterns, so that they neither overfit noisy data nor overlook subtle structures.

Our main insight is that memorization is not trivial but rather tied to the availability of underlying patterns in the data. It is suggested by the following key observations:

- **Non-triviality of Memorization Dynamics (§4.1):** We demonstrate that rather than steadily accumulating examples, transformer models repeatedly forget them (non-monotonicity), and the examples a model chooses to memorize at a given time are not simply those it saw most recently (non-recency).

- **Pattern-Dependent Memorization (§4.2):** We show that the propensity to memorize is highly dependent on the presence and types of patterns in the data. This reveals that what a model memorizes is tightly linked to the data's inherent patterns (or lack thereof).

- **Generalization and Pattern Acquisition (§4.3):** We find datasets where the model both memorizes and generalizes on exact sequence completion, and others where the model only memorizes despite the presence of patterns in the dataset. For the latter, we find some cases where patterns in the datasets have nevertheless been picked up by the memorizing model.

- **"Overthinking" in Larger Models (§4.4):** We identify a behavior in larger transformer models wherein they postpone memorization in settings with less-frequent patterns relative to smaller models. We hypothesize that the increased capacity can amplify the model's bias toward pattern-seeking, leading to late memorization of examples with fewer patterns.

## 2  Terminology

We provide definitions for some of our terminology in this section.

**Memorization**  There are many definitions of memorization [Zhang et al., 2023, Carlini et al., 2023c, Nasr et al., 2023b, Hartmann et al., 2023]. Broadly speaking, the difference lies in whether we have access to the training set or not. If we do, we can prompt with prefixes from training sequences and check for matching completions (*discoverable*). If we do not, we must construct prompts that can elicit completions corresponding to parts of training data (*extractable*).

Extractable memorization is relevant in the study of attacks that extract training data, which is beyond the scope of this study. Hence we focus on *discoverable memorization*:

**Discoverable Memorization (Nasr et al. [2023b]).** For a model Gen and an example $[p \parallel x]$ from the training set $X$, we say that $x$ is *discoverably memorized* if $\text{Gen}(p) = x$.

**Patterns**  By *pattern*, we refer to any quality of a dataset that could be exploited by a sufficiently expressive language model to more efficiently capture the data. This could range from simple

qualities such as containing a high frequency of specific trigrams to more intangible qualities such as containing only grammatically correct English sentences.

**Pattern Acquisition**   We say that a model has acquired or learned a pattern in our dataset if its predictions on new data adhere to that pattern, even if not necessarily being the overall correct prediction for that data. For example, in a dataset where all sequences are either uniformly lower or uppercase, and are lexicographically ordered (e.g., `aabbdfff`, `MMOSSST`), a model that continues `cc` with `a` has possibly acquired the case rule, but certainly not the lexicographic rule.

**Generalization**   Classically, a model is said to have *generalized* when it obtains better-than-random performance on unseen data drawn from the same distribution as the training set. This is tracked by testing performance on held-out validation and testing sets. For this work, we consider a stricter interpretation of generalization that requires successful predictions of multi-token sequence completions (similar to how we measure memorization). This allows us to align the two measurements.

We generally see good generalization as an indication of successful pattern acquisition, but don't necessarily expect the converse to be true.

## 3   Experimental Setup

In this section, we detail the data, models, and evaluation strategies for our study of memorization.

### 3.1   Datasets and Tokenization

**Natural Data**   We represent natural language with the help of samples from the WikiText-103 dataset [Merity et al., 2016], which we tokenize with the GPT-2 tokenizer. We build "natural-language" datasets of different sizes as subsets of this dataset, filtering always to samples with lengths from 100-300 tokens.

Example: `Troops are divided into five classes:  Scouts, Engineers, ...`

**Synthetic Data**   We also use the following synthetic datasets. For these, we maintain the same distribution of sequence lengths (100-300 tokens) and use a character-level tokenizer.

1. **In-Context Mapping (ICM)**: Have a fixed mapping of 10 letters to sequences of 3 digits. For each example, 3 letters (*L*) are selected. The example is then digit mappings of *L*, random sequence of letters in *L*, and digit mapping for the random letter sequence. Example: `A:123B:456C:789BBA456456123`

2. **Monotonic Increasing (MI)**: Next digit is previous + 1 mod 10. Example: `7890123...`

3. **Monotonic Increasing with Jumps (MIJ)**: Monotonic increasing with 15% chance of a random jump. Example: `234590123...`

4. **Monotonic Increasing with Jumps and Letters (MIJL)**: Similar to MIJ, but each jump is followed by a letter; letters also follow a monotonic increasing sequence. Example: `90126K78902L...`

5. **Shuffled WikiText**: Shuffle words in each WikiText sequence. Example: `into divided classes are Troops Engineers five :  Scouts, ...`

6. **Random Digits**: Randomly occurring digits. Example: `82671903...`

7. **Repeat 5 Tokens Ago**: Random sequences with 10% chance of repeating a token from 5 places ago. Example: `762130653...`

8. **Every 7th Token Same**: Either a random sequence, or (with 10% chance) a sequence where one of the first 7 tokens repeats every 7th token. Example: `1863054396...`

9. **Random Seed Lookup**: Each next digit is obtained by a lookup into the initial seed of length (*s*). Lookup index is the addition modulo *s* of last *m* digits. We use $s = 30$, $m = 8$. An example for $s = 5$, $m = 2$: `13820821...`

## 3.2 Selecting model configurations

We checked the model configurations of modern models and found that their architectures follow an approximately fixed model width ($E$) to model depth ($L$) ratio across scales. The GPT-2 family Radford et al. [2019] follows an $E/L$ ratio between 30 and 70; Llama 3 [Grattafiori et al., 2024] follows an $E/L$ ratio between 100 and 130. Following this insight, for a given parameter count, we select the models with an $E/L$ ratio in the range $[100, 130]$.

Moreover, scaling law papers either include the embedding layer in parameter count Hoffmann et al. [2022] or present the laws for both (including and excluding embedding layer) settings Kaplan et al. [2020]. The embedding layer constitutes a minute fraction of the total model capacity at real-world scales. We study much smaller models, where the embedding layer (which has size corresponding to the input vocabulary) constitutes 50-95% capacity of the entire model. Hence, we choose to exclude them from the parameter count when dealing with WikiText data. Our synthetic datasets use a much smaller vocabulary (less than 50 unique tokens) and the embedding layer need not be excluded. Therefore, the parameter counts henceforth refers to non-embedding parameters in the model.

## 3.3 Models and training

We use a decoder-only GPT-2 [Radford et al., 2019] architecture for our models. The model sizes studied are up to 12M non-embedding parameters; however, we do use GPT-2 small (85M non-embedding parameters) for a few experiments. The number of layers and the model's hidden dimension are determined as indicated above. The number of attention heads ($H$) is kept as $1$[1]. As stated, GPT-2 embeddings can be a very large fraction of model's capacity at small scales. Since we exclude them from parameter count, we must limit their contribution to model capacity while training. Hence, for experiments involving WikiText, we initialize the embedding layer of our models with that of the nearest (in parameter count) model from GPT-2 family and freeze it for the training. Principal Component Analysis (PCA) is used to reduce the dimensionality to the desired size. For experiments with synthetic data, the embedding layer is trained from scratch. Models are trained on datasets with varying number of examples (called 'ex' for short), using the language modeling (next-token-prediction) loss for 50k training steps with a batch size of 64. Adam optimizer is used with a learning rate of $1e - 3$. We train our models on NVIDIA A100 80 GB GPUs, and depending on the model and dataset sizes, the runs can take from 2 to 20 hours.

## 3.4 Evaluation

To evaluate memorization, we generate 30-token completions given a 50-token prompt from each sequence in the training data. We also measure performance on a held-out validation set when relevant (discussed in §4.3). For the completion, we track the following metrics over the course of training: (1) Exact match (EM), (2) Individual and Cumulative BLEU scores [Papineni et al., 2002], (3) METEOR [Banerjee and Lavie, 2005], (4) 13-gram Jaccard Similarity, (5) Training and Validation loss on the generated sequence. We note that all the metrics follow same trend, therefore, we choose EM as our primary metric and only show its trend for brevity. (See Appendices §A.3 and §A.4 for further clarification on these choices.) When EM over the training set is 1, the model has memorized its training data (up to our completion prompt length). For model selection purposes, we take an EM $\geq 0.95$ to denote memorization of training data. The smallest model that achieves this is said to be the *minimal model* for that dataset size. As an aside, we find that for full memorization, the *minimal model* size scales linearly with dataset size (see Appendix §C).

# 4 Experiments and Results

In this section, we present the main experiments from our study and the corresponding results on various properties of memorization. We present our findings in the order stated in §1. We first show that memorization is a non-trivial process, in that the set of memorized sequences is neither

---

[1]We validate our results on large models (Appendix §A.5), and with more attention heads (Appendix §A.6).

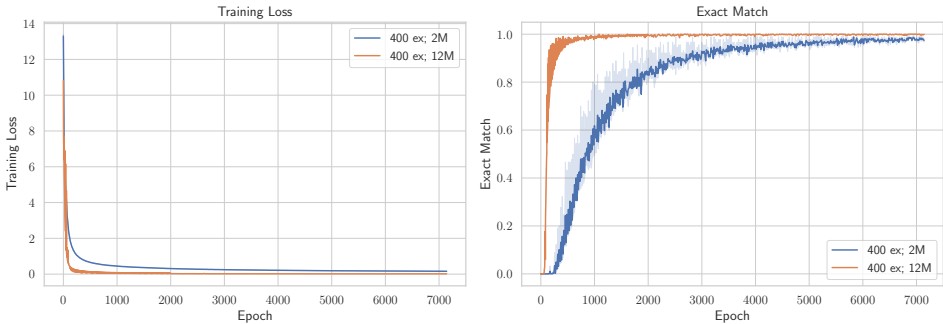

Figure 1: **Rate of memorization is not constant over training.** The rate of memorization as measured by exact match is shown for models of size 2M and 12M trained on a dataset of 400 examples. Memorization increases at a non-linear rate over training and the training loss decreases. (Error averaged over 3 independent runs. 95% confidence intervals over 1000 bootstrap trials.)

| 200 ex; 1M | 400 ex; 2M | 800 ex; 4M | 4000 ex; 12M |
|:---:|:---:|:---:|:---:|
| 29.4 | 20.5 | 19.3 | 39.4 |

Table 1: **Memorization of individual sequences is dynamic.** We track the memorization state of training sequences: memorized (EM = 1) or forgotten (EM = 0). Average number of times a sequence is forgotten for various model configurations is shown. Note that each sequence is memorized and forgotten many times.

constantly accumulated nor a cache of recently trained ones. Next, we show that memorization exhibits *differentiation* along sub-datasets with different patterns. We then verify that our models have *generalized* and not just memorized from some of our datasets, showing that this happens even when simple memorization is an option. Finally, for certain tasks, we observe a delay in memorization for larger models relative to small ones. We suspect that this phenomenon stems from failed attempts at pattern acquisition before memorization begins, and thus term it *overthinking*.

## 4.1 Temporal Non-Monotonicity and Non-Recency

**Result 1** Memorization is neither a constant accumulation of sequences nor recency-biased.

In Fig. 1, we show the training loss and EM for models trained on WikiText for 400 examples and two model sizes. We note that **the rate of memorization as denoted by EM is non-linear**. When the model has sufficient capacity for the dataset that it is trained on, **most of the dataset is memorized early on during training** (an EM score of 0.8 reached in under 50% of the training time required to reach an EM of 1). The sequences memorized at the end (or ones that were never memorized) tend to sometimes contain numbers (e.g., years). Note that we do not always observe such difference; yet, it suggests that the order of memorization may be related to the perceived difficulty or the amount of randomness in the sequence (which we touch upon in §4.2).

While the number of sequences memorized seem to increase over time (Fig. 1), the **memorization of individual sequences is dynamic**. To study the memorization of individual sequences, we track whether a sequence is memorized (EM = 1) or forgotten (EM = 0) by the model every 100 training steps during training. We find that individual sequences continue to be forgotten and re-memorized throughout training (Table 1). Using a 'softer' measure of forgetting (whether the sequence is among the top generations when sampling many times) also leads to the same conclusion (See Appendix §D.2). The above two insights denoting the non-linear rate and dynamism allow us to conclude that the memorization is not a constant accumulation of sequences (Result 1).

Since the model is not accumulating sequences, another plausible hypothesis is for memorization to be governed by the recency of sequences seen. In Table 2, we show the intersection between the number of newly memorized sequences and the recent batch as a fraction of both. We find that both of these are low across all scales, suggesting that the sequences a model chooses to memorize

|  | 200 ex, 1M | 400 ex, 2M | 800 ex, 4M | 1000 ex, 4M | 4000 ex, 12M |
|---|---|---|---|---|---|
| % of new mem. in last batch | 3.7 | 14.7 | 7.9 | 5.5 | 1.488 |
| % of last batch in new mem. | 6.54 | 4.37 | 4.2 | 10.0 | 8.32 |

Table 2: **Memorization is not governed by the recency of sequences.** We track the percentage of newly memorized sequences (whose EM changed from 0 to 1 in the last train step) from the last batch and vice versa. Both of these metrics are low across different scales.

|  | 100 ex; 0.5M | 200 ex; 1M | 400 ex; 2M | 800 ex; 4M | 2000 ex; 8M |
|---|---|---|---|---|---|
| Baseline | 0.96 | 0.96 | 0.96 | 0.98 | 0.95 |
| Shuffled | 0.68 | 0.63 | 0.70 | 0.69 | 0.72 |

Table 3: **Presence of patterns increases the memorization rate.** (Training on shuffled word order data reduces exact match.) We train different models and dataset sizes on both WikiText and Shuffled WikiText (where words in each sequence are randomly shuffled) for same amount of training time. A drop in EM of $\sim 30\%$ is observed for the latter, signifying that the linguistic patterns increase the rate of memorization.

at a particular training step are not necessarily the ones it has most recently seen in the last batch. Therefore, **recency of sequences does not govern memorization**.

## 4.2 Differentiation

> **Result 2** Models exhibit different memorization profiles for different datasets, both when trained independently on a single dataset and on data mixtures.

Next we ask: *Is all memorization alike (across datasets)?* We find the answer is *No*. We start by constructing a simple variant of the WikiText dataset: for each sequence, we shuffle the order of words randomly, leading to a new *shuffled* sequence. We find that models trained on this dataset for the same amount of time as original WikiText show a drop in EM by about 30% (Table 3). We also try other shuffle variants (Appendix §B.1) and have consistent findings. **This implies that (linguistic) patterns affect the memorization rate.**

To further investigate this finding, we design various datasets with different amounts of randomness and patterns (§3.1). We train models of size 4M on these datasets of size 500 examples. The EM curves for the models are plotted in Fig. 2a. We note that different datasets have different memorization profiles: with "Random Digits" on the far right and MI on far left. We also observe two distinct type of curves: *sudden jump* and *gradual rise*, which is another characteristic feature of the data the model is trained on. Both of these observations inform us that **patterns beyond *unigram statistics* are taken into consideration for memorization**.

In Appendix §B.2, we present more experiments to examine the effects of changing different hyperparameters in our synthetic datasets and make consistent observations.

Further, we check whether the same insight holds for models trained on data mixtures. We train models on two data mixtures: (a) mixture of monotonically increasing sequences with different amounts of randomness and random digits sequences, (b) WikiText mixed with random digits sequences. In both cases, we train three models each and plot the average metrics. The datasets have equal proportion of all component sub-datasets.

For (a), we train 4M-parameter models on dataset of 500 examples. We plot in Fig. 2b the number of epochs needed to achieve different levels of exact match for all component sub-datasets. We find that the model memorizes them in the order of gradual increase in the randomness from MI to "Random Digits". (For unbounded sequences, the probability of a jump in both MIJ and MIJL sequences would be the same. However, sequences in both of these datasets have fixed lengths and follow same length distribution. Therefore, MIJL is less random as the interleaved sequence of letters is deterministic after the first letter, making a larger portion of these sequences deterministic.)

For (b), we train 8M-parameter models on dataset with 2000 examples and plot the EM over time for both sub-datasets in Fig. 3 (Left). We observe that memorization happens in phases: almost all of WikiText is memorized before the memorization of random sequences starts. This agrees with

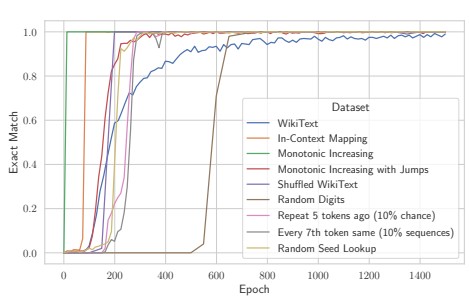
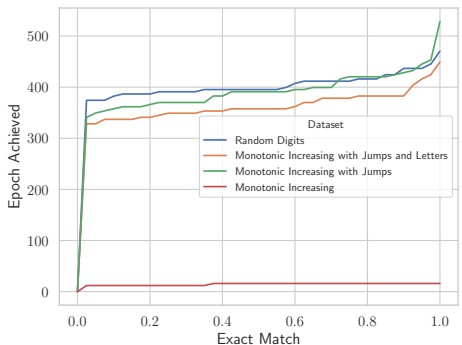

(a) Trained on individual datasets         (b) Trained on a mixture of datasets

Figure 2: **Models exhibit different memorization profiles and the order of memorization for different datasets when trained on them independently (left) or simultaneously (right).** *Left:* The exact match for various datasets for models of size 4M trained on 500 examples. *Right:* Number of epochs required to achieve different exact match values for each sub-dataset are shown for a model of size 4M trained on 500 examples equally distributed across the four different datasets (MI, MIJ, MIJL, and Random Digits). Note that the datasets with more randomness take longer to be memorized.

the insight in §4.1 where we noted sequences of numbers preferentially being memorized at the end. Moreover, in Fig. 3 (Right), we plot overall (combined) EM and training loss for an 8M model trained on 2000 examples for different mixture compositions of WikiText and random digits datasets. The phase changes in loss and EM are evident. Note the position at which the phase corresponding to memorization of random digits starts: the less the amount of random digits data, the later their memorization starts. These phase shifts are reminiscent of disjoint, yet qualitatively similar results in Chen et al. [2025], where they note sudden drops in loss for syntax acquisition.

These observations suggest that **the order and speed of memorization of different datasets are related to the extent and type of patterns contained in them**, leading to different memorization profiles.

## 4.3 Generalization and Pattern Acquisition

> **Result 3** Models generalize fixed mapping-based patterns on unseen data.

*Do models generalize these patterns to an unseen validation set?* For this, we test our models on datasets with deterministic patterns such as MI and ICM on a disjoint validation set. Since there are only 10 unique monotonically increasing sequences (one each with a different starting digit), to test generalization, we held out the sequence starting with digit '7'. Models of different sizes trained on different data sizes were able to successfully complete sequences starting with '7'. For ICM dataset, the "500 ex; 4M" model when evaluated on an unseen validation set of 500 examples, achieves an EM = 1. The underlying letter-to-digits mapping is same across all sequences in train and validation sets. Note that both of these datasets have sequences with static lookup patterns. In MI, the next digit is deterministic based on the previous digit (bigram lookup), while the ICM sequences require lookup from a fixed letter-to-digits mapping by definition. This implies that **the models can generalize to datasets with static lookups**. The patterns in these datasets can therefore be acquired, and hence, the model uses reconstruction to memorize them.

To test the extent of generalization of lookup-based patterns, we test on "Random Seed Lookup" dataset. We also design a *dynamic* version of the ICM dataset where the letter-to-digits mapping is dynamic across sequences. We train 4M-parameter models on 500 examples each from these datasets. While the models memorize these datasets perfectly (EM = 1), the EM on the validation set remains identically zero. Therefore, we find that **for more complicated and dynamic lookups, the models fail to generalize**. So, we suspect that this pattern was not acquired by the model, and the performance on training set corresponded to the recitation of those examples.

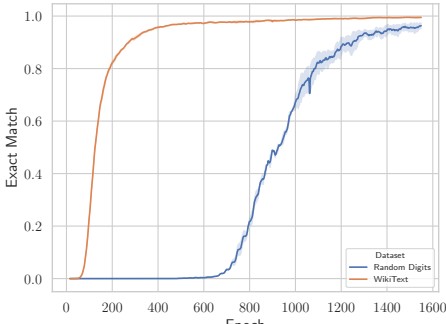 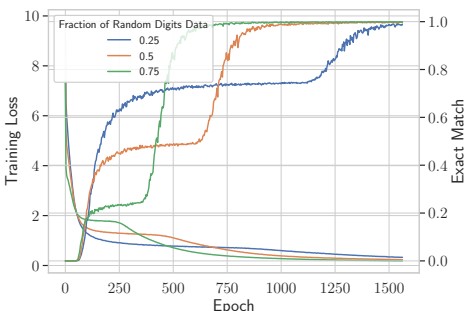

Figure 3: **Phase shifts corresponding to the memorization of a mixture of WikiText and Random Digits data.** *Left:* EM for the two sub-datasets for an 8M model trained on 2000 examples (an equal mixture of WikiText and random digits). Memorization of random sequences start after most of WikiText portion has been memorized. *Right:* EM for all training sequences (all sub-datasets combined) for "2000 ex; 8M" models with different fractions of random data. The length of each phase is proportional to the amount of the corresponding sub-data.

*But what about generalization in datasets that partially contain deterministic patterns with some amount of randomness?* To answer this question, we check how often do the models trained on MIJ dataset follow the *monotonically increasing* pattern on an unseen validation set. Given 50-token prompts from sequences in this validation set, we generated 30-token completions for 4M and 8M models (both trained on 500 sequences). For every bigram $(x_{t-1}, x_t)$ in completions, we measured how often is it monotonically increasing (i.e., $x_t = (x_{t-1} + 1)\%10$). We find this number to be 97.3% and 98.1% for the two models respectively. This is much larger than the random chance (10%) of generating a monotonically increasing next digit. Therefore, despite there being randomness (15% random jumps), the model acquires the monotonically increasing pattern.

**These results demonstrate models' affinity toward patterns for memorization, and the generalization of static lookup-based patterns to unseen sequences.**

### 4.4 Overthinking

> **Result 4** (Overthinking): On datasets with weak or absent patterns, large models memorize slower than smaller models.

While investigating our central thesis (pattern-seeking behavior in memorization), we made an interesting observation: on certain datasets without patterns or with less-frequent patterns, the memorization in larger models can be delayed relative to small models. We believe it is caused by the models' inherent affinity to patterns, which potentially backfires when patterns cannot be found.

In Fig. 4, we plot the EM for a small and a larger model trained on 500 examples each from different datasets. The model sizes are mostly 4M and 8M respectively, with the exception of dataset with *weak* patterns (repeat some previous token with low probability), where the larger model is GPT-2 small [2]. We note that for dataset with abundant patterns (linguistic patterns such as in WikiText, or static mapping-based ones for ICM and MI family), the larger model is either better or on par with small model. However, for models with large amounts of randomness (weak pattern datasets, "Random Digits"), or complex dynamic lookups ("Random Seed Lookup"), larger models have a much worse memorization rate. Therefore, **a larger model is not always better or faster**. We call this behavior *overthinking*, and hypothesize its cause being **model's pattern-seeking behavior which may lead to a delay when patterns are missing or not present in abundance.**

---

[2]For weak patterns, the effect was only evident at a higher model scale.

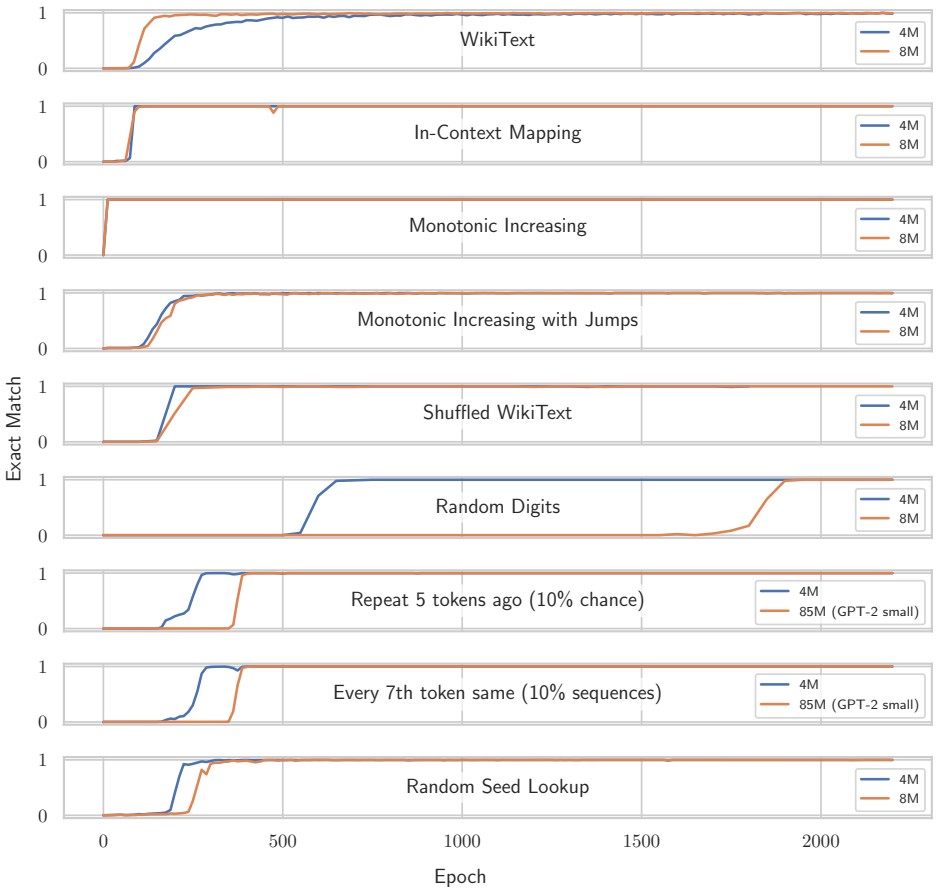

Figure 4: **Overthinking.** EM for a small and larger model trained on dataset of size 500 examples is shown across many datasets. Larger model is slower for data with absent (Random Digits) and weak (repeating a previous token with low probability) patterns, and for complex dynamic lookups.

## 5 Related Work

Memorization has been a central topic in machine learning since the earliest neural networks. We organize prior work into six themes: (1) classical and associative memory models, (2) memorization in modern language models, (3) privacy and data extraction, (4) memorization versus generalization and pattern-seeking behavior, (5) interpretability and training dynamics, and (6) related learning phenomena such as forgetting and curricula.

**Classical and Associative Memory Models**  The study of neural associative memories began with Hopfield networks Hopfield [1982], which showed that recurrent networks could store discrete patterns as stable attractors. These systems, along with bidirectional associative memories Kosko [1988] and sparse distributed memory Kanerva [1988], were designed to explicitly encode and retrieve stored patterns from the network's weights. Recent surveys emphasize that modern LLMs behave differently from these classical models, exhibiting implicit memory through distributed representations, attention mechanisms, and in-context learning rather than explicit associative storage Zhang et al. [2024]. This shift motivates a fresh look at memorization in the transformer era.

**Memorization in Transformer Language Models**  Large-scale studies demonstrate that transformers memorize exact sequences from their training data, starting with early extraction experiments on GPT-2 Carlini and Trumpler [2021]. Subsequent work confirmed that unique or duplicated examples can be reproduced verbatim Carlini et al. [2023b], Nasr et al. [2023a], Jagielski et al. [2022], Ippolito et al. [2022]. Empirical evidence indicates that memorization grows with model scale, dataset du-

plication, and training length; strategies such as deduplicating training data mitigate this tendency Lee et al. [2021]. Alternative framings include counterfactual memorization Zhang et al. [2021] and adversarial compression Schwarzschild et al. [2024]. Memorization is also observable across modalities, including diffusion models Carlini et al. [2023a], code models Yang et al. [2023], and vision models Lukasik et al. [2023].

**Privacy and Data Extraction**    Memorization poses privacy and IP risks: it can lead to unintended data leakage, including PII and copyrighted text Carlini and Trumpler [2021], Carlini et al. [2022, 2023b], Nasr et al. [2023a], Ippolito et al. [2022]. Large-scale attacks show thousands of training examples can be extracted cheaply Nasr et al. [2023a], and model-stealing setups remain vulnerable even with API-only access Carlini et al. [2024].

**Memorization vs. Generalization and Pattern-Seeking Behavior**    While memorization is often viewed as the opposite of generalization, research shows that networks can memorize random labels Zhang et al. [2017], yet tend to learn simpler patterns first before memorizing irregularities Arpit et al. [2017b]. The inductive bias to capture structure first is evident in language models through frequency-driven rule acquisition Wei et al. [2021], memorization-generalization continua Dankers et al. [2023], and long-tail memorization dynamics McCoy et al. [2020]. Taxonomies such as recitation vs. reconstruction highlight multiple modes of memorization Prashanth et al. [2024]. Grokking shows sudden generalization after overfitting Power and et al. [2022]; notably, memory can serve as a form of compression Schwarzschild et al. [2024]. Our results align with these dynamics.

**Interpretability and Training Dynamics**    Training-dynamics studies highlight phase transitions where models abruptly acquire structural knowledge Chen et al. [2023]. Representation analyses reveal that linguistic features emerge in progressive layers Hewitt and Manning [2019]. Beyond training, methods for updating models with knowledge explore the interplay between memorization and reasoning post hoc Li and Goyal [2025].

**Related Learning Phenomena: Forgetting and Curricula**    Early work on catastrophic forgetting documents how models overwrite prior knowledge McCloskey and Cohen [1989], Ratcliff [1990]. This persists in modern models, where fine-tuning can cause forgetting of memorized examples Jagielski et al. [2022]. Continual memorization research aims to avoid this trade-off Chen et al. [2024]. Curriculum learning shows that ordering training from easy to hard promotes generalization over rote memorization Bengio et al. [2009]. Our observations suggest a natural analog: transformers tend to memorize in ordered bursts aligned with data patterns.

# 6    Limitations

Our study is conducted on relatively small models under a multi-epoch training regime, which does not directly reflect the large-scale, single-epoch training used for modern foundation models, but does provide relevant insights. Additionally, while synthetic datasets can be designed with the intent of diversity in type and "difficulty" of patterns in mind, identifying and quantifying patterns in real-world language modeling data (e.g., WikiText) remains challenging. Finally, our evaluation of memorization is based on 30-token completions, which may not capture subtler forms of memorization or longer-span dependencies.

# 7    Conclusion

In this work, we studied how memorization arises in transformer language models through the lens of training dynamics. We showed that memorization is not a monotonic or recency-based accumulation, but a pattern-driven process: models memorize in bursts, generalize when possible, and delay memorization when patterns are ambiguous. Larger models often "overthink" low-pattern data, suggesting a strong inductive bias toward structure. These findings suggest that memorization and generalization emerge from a shared mechanism of pattern-seeking. Several questions remain open. What kinds of patterns are most conducive to memorization? Can we impose a hierarchy over them, and understand how the model prioritizes one over another during training? Another direction is to understand the mechanistic underpinnings of overthinking, and whether this behavior can be encouraged or mitigated based on downstream goals.

## 8 Acknowledgements

We are grateful to Deniz Bayazit and Badr AlKhamissi for their helpful discussions during this project, and for their feedback on our manuscript. We also gratefully acknowledge the support of the Swiss National Science Foundation (No. 215390), Innosuisse (PFFS-21-29), the EPFL Center for Imaging, Sony Group Corporation, and a Meta LLM Evaluation Research Grant.

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

# Contents

# A   Experimental Setup Details

## A.1   Examples from WikiText and Sequence Length Distribution

Some examples from the WikiText dataset are shown in Table 4. As mentioned in §3.1, we filter the dataset based on the length of sequences The resulting distribution of lengths is plotted in Fig. 5. For consistency, we maintain the same sequence length distribution across our synthetic datasets.

| Example | Text |
|---|---|
| Example 1 | Cicely Mary Barker ( 28 June 1895 - 16 February 1973 ) was an English illustrator best known for a series of fantasy illustrations depicting fairies and flowers .  Barker 's art education began in girlhood with correspondence courses and instruction at the Croydon School of Art .  Her earliest professional work included greeting cards and juvenile magazine illustrations , and her first book , Flower Fairies of the Spring , was published in 1923 ... |
| Example 2 | The Gregorian Tower ( Italian :  Torre Gregoriana ) or Tower of the Winds ( Italian :  Torre dei Venti ) is a round tower located above the Gallery of Maps , which connects the Villa Belvedere with the Apostolic Palace in Vatican City .  The tower was built between 1578 and 1580 to a design by the Bolognese architect Ottaviano Mascherino ( who was credited with building the Apostolic Palace ) mainly to promote the study of astronomy for the Gregorian Calendar Reform which was commissioned by Pope Gregory XIII and promulgated in 1582 .  It was then also known as the Tower of Winds .  The tower is now called the " Specola Astronomica Vaticana " , the Vatican Observatory ... |

Table 4: Some examples from the WikiText dataset.

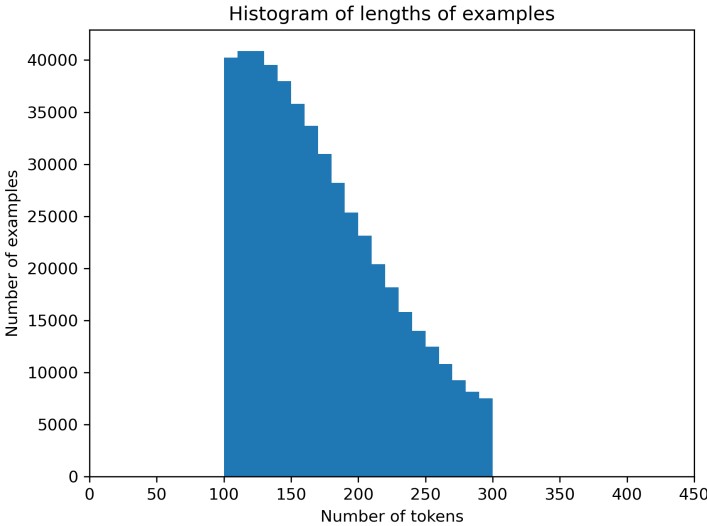

Figure 5: Distribution of sequence lengths in the WikiText dataset after filtering.

## A.2 Selecting Model Configurations

Our model configurations follow the model width-over-depth ($E/L$) ratio from Llama 3 [Grattafiori et al., 2024] family of models. We plot $E$-$L$ contours corresponding to non-embedding parameters for each desired parameter count. Next, we plot the lines corresponding to different, fixed $E/L$ values. We then choose the model configurations nearest to the intersection between each contour and these lines. The contours, $E/L$ lines and the selected model configurations are shown in Fig. 6.

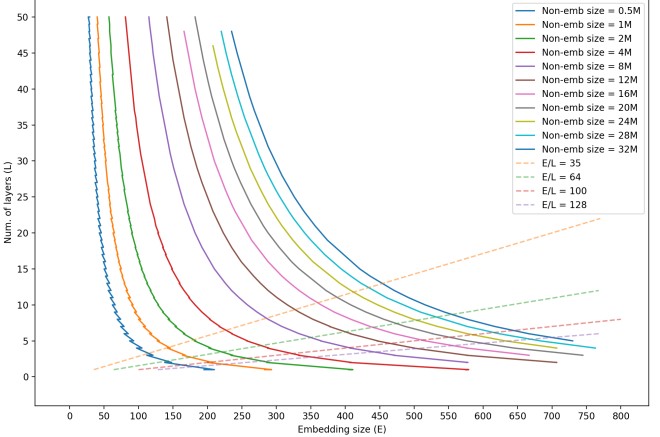

| # Params | $L$ | $E$ |
|----------|-----|-----|
| 0.5M | 2 | 144 |
| 1M | 2 | 204 |
| 2M | 3 | 235 |
| 4M | 3 | 333 |
| 8M | 4 | 408 |
| 12M | 4 | 500 |
| 20M | 5 | 577 |
| 32M | 6 | 666 |

Figure 6: **Selecting model configurations.** *Left:* To select the model configurations for different model sizes, we choose model width ($E$) and number of layers ($L$) such that the $E/L$ ratio is between 100 and 130 (following Llama 3), i.e., models lying at the intersection of contours and lines. The number of heads ($H$) is 1 in all chosen configurations. *Right:* Chosen $E$ and $L$ values across different model sizes.

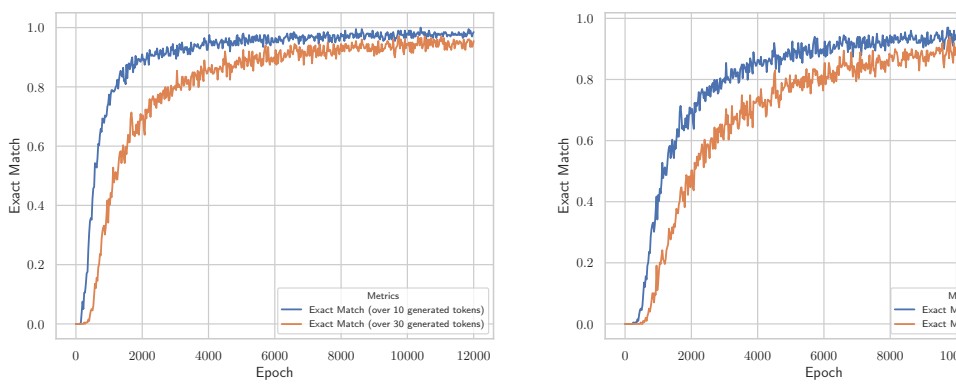

Figure 7: **Ablation on the prompt and generation lengths used for evaluation for "200 ex; 1M" model.** *Left:* Prompt length is 50 tokens and results for generation lengths of 10 and 30 are shown. *Right:* Two prompt and generation length combinations are shown: the default (50-30) and (30-70). Note that the trends still continue to be the same across all these choices.

## A.3 Choice of Prompt and Generation Lengths

In §3.4, we stated our choice of evaluating different memorization metrics on 30-token completions given a 50-token prompt. We now show that these specific choices do not fundamentally change the results we observe. In Fig. 7, we plot EM in two scenarios: (a) keeping the prompt length as

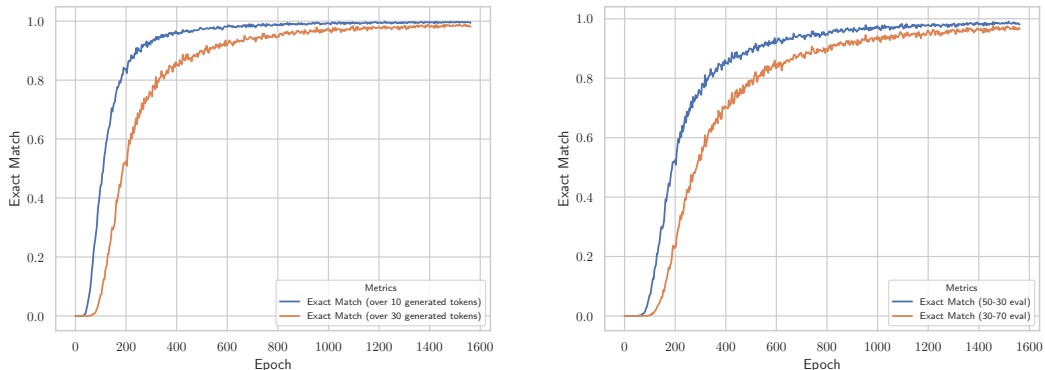

Figure 8: **Ablation on the prompt and generation lengths used for evaluation for "2000 ex; 8M" model.** *Left:* Prompt length is 50 tokens and results for generation lengths of 10 and 30 are shown. *Right:* Two prompt and generation length combinations are shown: the default (50-30) and (30-70). Note that the trends still continue to be the same across all these choices.

50 tokens and plotting results for generation lengths of 10 and 30; (b) varying both the prompt and generation lengths. We try two configurations: (50-30) and (30-70). Fig. 8 shows the same plots for "2000 ex; 8M" model. The trends still continue to be the same across all these choices.

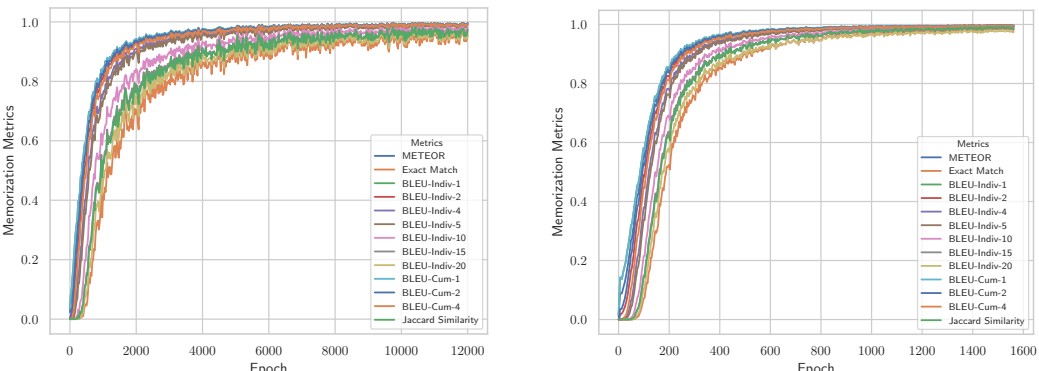

Figure 9: **Different metrics for memorization.** Many metrics are plotted to track memorization for "200 ex; 1M" (Left) and "2000 ex; 8M" model (Right). Note that all the metrics follow the same trend.

## A.4    Other Metrics for Memorization

In Fig. 9, we plot different metrics we track for memorization for "200 ex; 1M" and "2000 ex; 8M" models trained on WikiText. All metrics we tracked follow the same trend and hence we choose to only show EM for brevity.

## A.5    Model Sizes

We experiment with larger models to show evidence for the applicability of our findings to larger model scales. We replicated our setup on model sizes 100M, 200M, 500M, and 1B, on datasets "Every 7th token same (10% sequences)" and "Monotonic Increasing with Jumps." We consistently observe that larger models take longer (aka 'overthinking') before starting to memorize sequences in these datasets (same as Fig. 4). Please note that these training runs demonstrating overthinking can be very computationally expensive for larger models, as they take much longer to start memorizing those datasets, limiting the model sizes we can include in the study. However, these results up to 1B scale conclude that overthinking does hold at large scales.

We also trained 200M and 500M models on the following datasets (one at a time): "Monotonic Increasing", "Monotonic Increasing with Jumps", "In-Context Mapping", and "Random Digits". We observed different memorization profiles (same as Fig. 2a), with "Random Digits" sequences being memorized last.

Hence, differentiation and overthinking are both applicable at larger model scales.

## A.6 Attention Heads

In our initial experiments, we observed that the number of attention heads does not impact the results, and hence we chose to use a single attention head. We did more experiments with models having 12 attention heads and found that our results still hold. Specifically, we train a 4M-parameter model having 12 attention heads on 500 examples each from "Monotonic Increasing" and "Monotonic Increasing with Jumps" datasets. We observed different memorization profiles (same as Fig. 2a). We also trained models with 4M and 8M parameters on 500 examples of "Random Digits" data and noticed delayed memorization in the 8M model (like Fig. 4). Hence the main results of our study ('differentiation' and 'overthinking') are not impacted by a change in the number of attention heads.

# B Dataset ablations

## B.1 Controlling the Degree of Shuffed-ness

In Table 3, we show that the memorization of the dataset drops by 30% when the WikiText data is shuffled at the word level. We also check the effect of varying the degree of shuffled-ness between the original and word-level shuffled states. We shuffle at the sentence and multi-sentence levels. Moreover, we try a shuffle variant where we preserve the 2- and 3-grams in the top 50th percentile (in terms of occurrence counts) as one unit and shuffle them with the rest of the sequence. This lets us control the extent of patterns in the dataset. We present the 'exact match' results on 0.5M (trained on 100 examples) and 1M (trained on 200 examples) models in Table 5. Consistent with our claims, we observe that the memorization capacity of a model is dependent on the extent of exploitable structure present in the data. With reduced structure (increased degree of shuffled-ness), the model memorizes less data.

| Shuffle variant | 100 ex; 0.5M | 200 ex; 1M |
|---|---|---|
| Baseline (original WikiText) | 0.96 | 0.96 |
| Shuffle 2-sentence units | 0.94 | 0.95 |
| Shuffle sentences | 0.93 | 0.92 |
| Preserve top 50% 3-grams | 0.78 | 0.78 |
| Preserve top 50% 2-grams | 0.72 | 0.73 |
| Word-level shuffled | 0.68 | 0.63 |

Table 5: Effect of different shuffling strategies on memorization (exact match scores are shown).

## B.2 Further Studies with Synthetic Datasets

To understand memorization dynamics more clearly, we do some more in-depth studies with our synthetic datasets by varying the different dataset hyperparameters. We present the results from these experiments with 4M models trained on 500 examples from the indicated datasets.

1. **Vary the jump probability in MIJ dataset.** We check for the following jump probabilities (in %): 10, 15, 20, 40, 50. As the probability of jumps increases (thereby increasing randomness), it takes longer to memorize the dataset, which is consistent with our claims.

2. **Use 2 or 3 letters instead of just 1 after each jump for MIJL dataset.** When more letters are used after the jump, memorization is faster. As the sequences are of finite length, with an increased number of monotonically increasing letters, the randomness decreases, making the sequences easier to memorize.

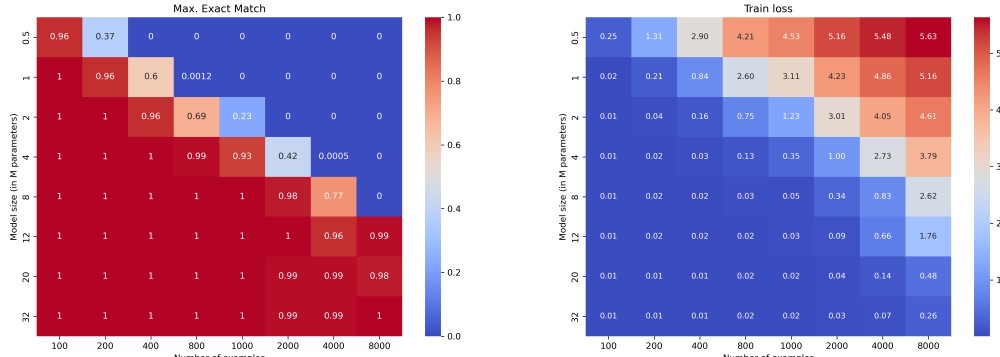

Figure 10: **Minimal model size for memorization scales linearly with dataset size.** Maximum Exact Match and final training loss across different model and dataset sizes are plotted. When models have enough capacity to memorize the dataset, the exact match is close to 1 (red region). When the models lack sufficient capacity, the exact match is close to 0 (blue region). Minimal models for each dataset size lie at the boundary of these two regions.

3. **Check for 4/5/6/8 tokens ago as in the "Repeat 5 Tokens Ago" dataset.** Exact Match curves for all these runs (4/5/6/8 tokens ago) jump from 0 to 1 in the range of [230, 280] epochs without any particular order. This denotes that there is no significant difference in the difficulty of these datasets in terms of memorizability.

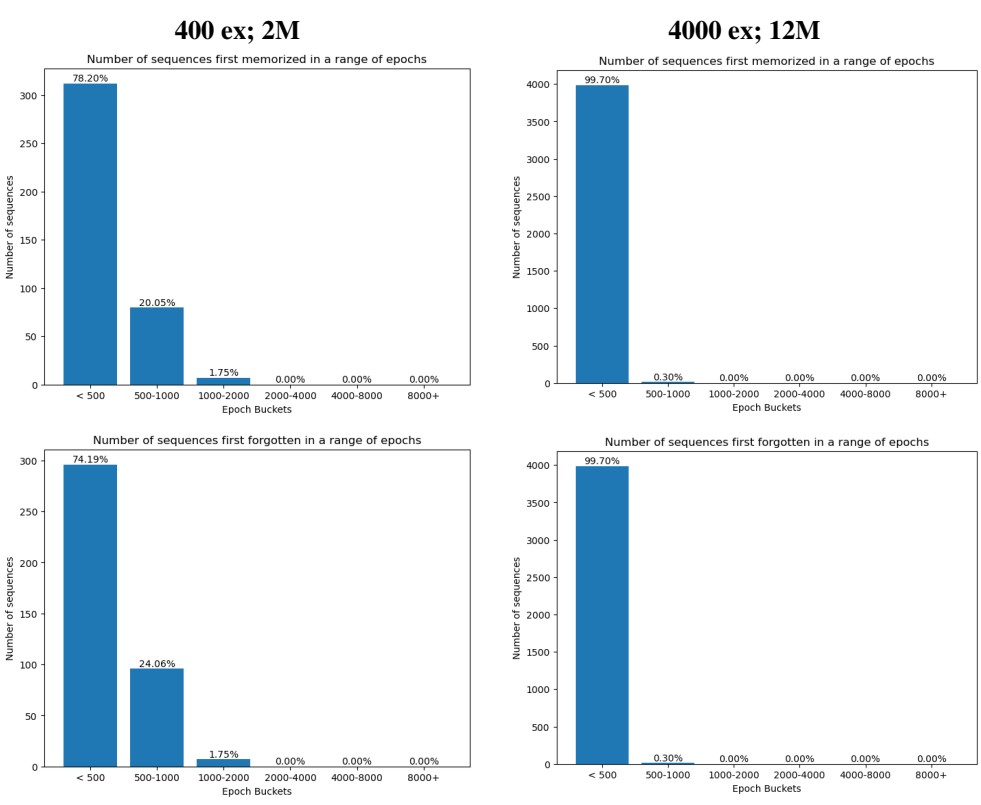

Figure 11: **First Memorized and First Forgotten statistics for models trained on WikiText.** First row denotes the model configurations. The percentage of total sequences that make up each bar is also written on top of it. We note that models memorize most sequences for the first time early on; larger models do it faster. Also, first forgotten and first memorized statistics are similar.

## C  Note on Scaling

While the main subject of our study is the training dynamics of memorization, we make an interesting observation regarding how it scales. We trained models of different sizes on datasets of different sizes until the training loss saturated, and found that the size of the smallest model needed to memorize the training data (EM $\geq 0.95$) scales linearly with the dataset size (Fig. 10).

## D  Further Dynamics of Memorization

### D.1  First-Memorized and First-Forgotten Statistics

In §4.1, we saw that memorization is dynamic and sequences switch between *memorized* and *forgotten* states. We also track the epoch when a sequence is first memorized and first forgotten. These two distributions are plotted in Fig. 11 for "400 ex; 2M" and "4000 ex; 12M" configurations trained on WikiText. We make two observations: (a) models do prefer to (first) memorize most sequences early on, with larger models (first) memorizing in fewer epochs; (b) first-memorized and first-forgotten distributions are similar. We also plot these metrics for mixtures of WikiText and Random Digits datasets for "2000 ex; 8M" model in Fig. 12, and make same observations. Additionally, we observe the two distinct phases (as in Fig. 3) corresponding to the memorization of WikiText and random data.

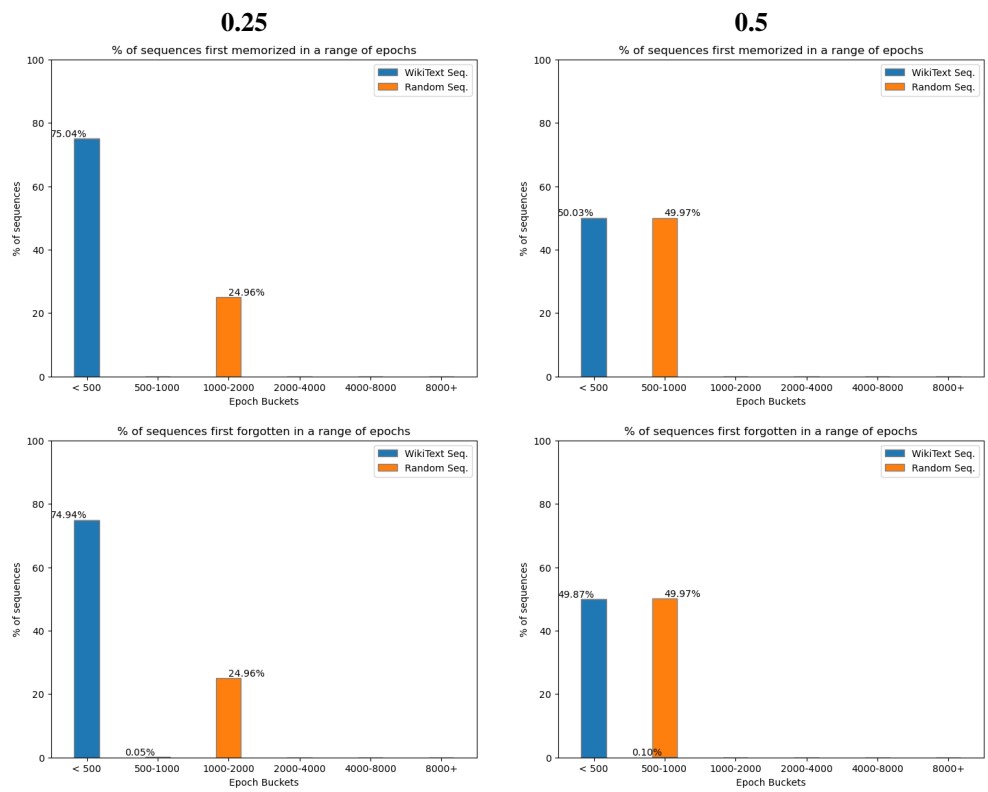

Figure 12: **First Memorized and First Forgotten statistics for 8M-parameter models trained on a mixture of WikiText and Random Digits data.** Topmost row denotes the fraction of random data in the mixture. The percentage of sequences (out of 2000) memorized across epochs is plotted. WikiText sequences are (first) memorized prior to Random Digits sequences.

### D.2  Using Softer Metrics to Measure the Dynamic Nature of Memorization

We showed that sequences continue to be forgotten and re-memorized throughout training (i.e., memorization is dynamic) by using exact match as the metric. Here, we check if the same observation

| Epochs | Topics |
|---|---|
| $< 500$ | Topic 0:  for to and they that males be it are episode
Topic 1:  the was in by and to for of on it
Topic 2:  the of and in to was on as that with
Topic 3:  of is her and an she for the as to
Topic 4:  is and to his it at that with he on |
| $500 - 2000$ | Topic 0:  16 1961 21 22 24 27 28 29 31 36
Topic 1:  16 1961 21 22 24 27 28 29 31 36
Topic 2:  16 1961 21 22 24 27 28 29 31 36
Topic 3:  the of and size castles distinctive has great geography for
Topic 4:  16 1961 21 22 24 27 28 29 31 36 |

Table 6: Topic modeling results (top 5 topics) on sequences memorized at different points during training for the "2000 ex; 8M" model.

| S. No. | Sequence |
|---|---|
| 1. | 1915 - 16, 1917 - 18, 1918 - 19, 1920 - 21, 1921 - 22, 1923 - 24, 1926 - 27, 1927 - 28, 1928 - 29, 1930 - 31, 1935 - 36, 1936 - 37, 1948 - 49, 1949 - 50, 1950 - 51, 1955 - 56, 1957 - 58, 1959 - 60, 1961 - 62, 1963 - 64, |
| 2. | The size of these castles varied depending on the geography of the site, the decisions of the builder and the available resources.  Analysis of the size of mottes has shown some distinctive regional variation ; East Anglia, for example, saw much larger mottes being built than the Midlands or London.  while motte @-@ and @-@ bailey and ringwork castles took great effort to |

Table 7: Sequences not memorized by a "4000 ex; 12M" model.

holds under a "softer" metric. We track the fraction of times a sequence is sampled (out of 5 generations) with a temperature of $0.5$. We call this sampling accuracy ($s_{acc}$). We also track the status of each sequence at a given timestep: 0 (forgotten), 1 (memorized), as found via exact match. We report our findings on the configurations: "100 ex; 0.5M", "200 ex; 1M", and "400 ex; 2M".

We average status and $s_{acc}$ across sequences and note that their trend across epochs is similar to Fig. 1 (rising from 0 to 1) and that they almost overlap. This suggests that the assignment of sequence status using exact match agrees with this softer sampling accuracy metric.

The average $s_{acc}$ across the sequences with status 0 (forgotten) was about $10 - 15\%$, while the average $s_{acc}$ across the sequences with status 1 (memorized) was about $99\%$. This strongly suggests that sequences that tend to be forgotten, as detected by exact match, also tend to have a low chance of being generated by the model.

Also, we use exact match since softer measures are not relevant in the context of PII or copyright infringement, which are real-world memorization issues.

### D.3   Sequences Memorized across Different Epochs

To understand what type of sequences are memorized at different points during training, we perform topic modeling on different epoch buckets. The topics so obtained for the "2000 ex; 8M" model are shown in Table 6. Note that the topics for epochs $> 500$ primarily have numbers. Moreover, we observe that the sequences that are never memorized also sometimes happen to be abundant in numbers. In Table 7, we show such sequences for the "4000 ex; 12M" model.

## D.4 Metrics for Non-Memorized Sequences

We also track memorization metrics for sequences that are not memorized (EM = 0) at any given point during training. We show three such metrics in Fig. 13 for "400 ex; 2M" model, and make two observations: (a) Initially, when nothing is memorized, the model gradually memorizes parts of each sequence; (b) Later on, as more sequences get memorized, the ones that are being forgotten still have most of their parts memorized.

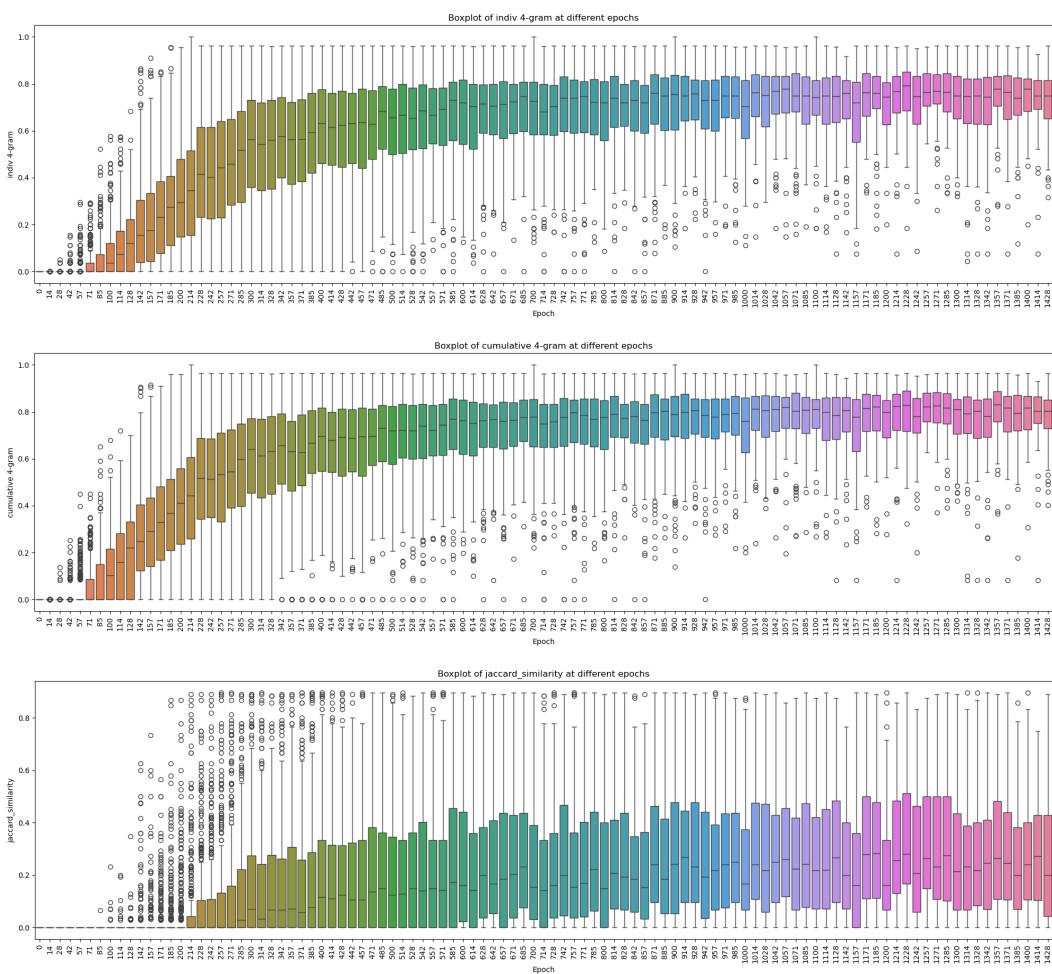

Figure 13: **Metrics for non-memorized sequences.** Individual and Cumulative 4-gram BLEU, and 13-gram Jaccard Similarity are shown for "400 ex; 2M" model. Note that all metrics gradually rise until saturation.

