# OpenReview forum: "For Better or for Worse, Transformers Seek Patterns for Memorization"
_NeurIPS.cc/2025/Conference — NeurIPS 2025 poster_

### Official Review · Reviewer_eqVj · 2025-06-22

**Clarity:** 1
**Significance:** 2
**Originality:** 2
**Rating:** 3
**Confidence:** 4

**Summary:**

This paper studies the question “What affects which facts are memorized by a language model during pretraining and when/at what rate they are memorized?” The study is performed on small GPT-2 models trained on subsets of Wikitext, on small synthetic datasets containing, e.g., random digits, and combinations of both. The authors find that (1a) strings continue to be forgotten and re-memorized throughout training (1b) It is not always the most recent strings that are “freshly” memorized at any moment (2) subsets of the data that follow clearer patterns are memorized earlier, with clear phase shifts when training on mixture datasets (3) larger models delay memorization of strings with no patterns, which the authors speculate is because these models try to seek patterns to learn before memorizing a given sequence.

**Questions:**

- As mentioned in section 2 of Weaknesses, do one or both of the two metrics I suggested (or any others that measure a “softer” sense of forgetting) show that sequences are indeed forgotten and re-learned all the time?
- Could you present more evidence or arguments as to why the results of section 4.4 should be interpreted as the model seeking patterns (as opposed to the larger model learning facts more slowly for some other reason)?
- Could you clarify the use of the terms “pattern” and “generalization”? Ideally, a more formal definition would be welcome, as that would help delineate what results should be more or less surprising.
- Out of all the results that are presented in the paper, which one would you say is the central contribution of your paper, and the one that summarizes the paper the best?

**Ethical Concerns:**

["NO or VERY MINOR ethics concerns only"]

**Final Justification:**

I will paste the response to the authors here, as I think it summarizes the justification well.
- I am glad to see an expansive list of related work. This addresses that concern.
- The clarification of the narrative is welcome, and I would encourage the authors to also apply this to streamline their paper further.
- I acknowledge that the authors do not wish to anthropomorphize the model. Yet, some readers and especially future work might mistakenly do so upon reading this paper, so I suggest that a different word than "seeking" be used when describing the delayed onset of learning, if possible. However, this is just a minor technical point.
- The expansion of results to models up to 1B also greatly strengthens the paper, as do the results with soft metrics. I will still contend that using 100-1000 epochs is rather excessive and may skew takeaways.

Overall, I find that the rebuttal is reasonably convincing. I will therefore raise my score to "3: borderline reject" keeping both the improvements and the lingering concerns in mind.

**Limitations:**

yes

**Quality:**

2

**Strengths And Weaknesses:**

**Strengths**

- From a scientific standpoint, I find the questions raised at the beginning of the paper very intriguing—specifically, what dictates which facts are memorized by a language model and how fast?
- Some of the results here are surprising (at least to me). For example, the fact that a larger model might take longer to memorize some sequences is interesting, and to my knowledge, novel.
- The setup is described clearly, and the description of the experiments is succinct but informative. Throughout, I had no trouble understanding what the current experiment was trying to do.

**Weaknesses**

- I don’t think that the setup adopted here is a faithful description of model pretraining. For example, Figure 1 trains on 400 examples for over 7000 epochs! In reality, we have millions or billions of examples for 1-4 epochs. Several of these results only seem to hold in the regime of 300+ epochs.
- The claim that sequences are constantly forgotten and relearned warrants further study. If I am correct, the authors use greedy decoding (n=1) and use Exact Match to check if a sequence has been forgotten. It would be more suitable to instead measure if a sequence remained among the top few completions preferred by the model, measurable using either perplexity or the fraction of times the exact sequence was sampled when sampling several times with a temperature. I suspect that with these metrics, sequences will appear to have been forgotten much fewer times throughout training.
- The word “pattern” is used rather loosely throughout the paper, which renders some of the takeaways ambiguous. I’ll try to give an example: the phrase “In the year…” might appear several times in this same order in the corpus, but I would not consider this trigram a “pattern”. In that case, I don’t find the results of Table 3 surprising - these sequences of words that always appear in the same order get jumbled when you shuffle the sequences, and hence, you get sequences that are harder to memorize. These issues might go away with a more rigorous definition of “pattern” and “generalization.”
- Result 4 in the paper anthropomorphizes the language model, which leads to a leap in conclusions. For one, I think it would take much more evidence to make the case that (larger models memorize random facts slower) implies (larger models seek patterns to learn and hence take longer to memorize random facts). Besides, I don’t think any of this involves any “thinking” on the part of the model, because this describes the training process, not inference. Therefore, I find the term “overthinking” misleading.
- More generally, I think the paper does not have a very clear direction in its current form. I will concede that it opens with an interesting question and has some very interesting results, but the results do not come together to tell a story or describe any particular phenomenon. In particular, I think the section on generalization to static v/s dynamic patterns does not fit well into a paper that is interested in understanding memorization.
- While I don’t want to play the role of citation police too much, I do think that this paper needs a much more expansive study of related work—the related work section has no citations from before 2023, except for a 2017 paper. The problem of memorization has been studied for a very long time—as far back as 1989 (see [1] and [2] as examples).

[1] McCloskey, M., & Cohen, N. J. (1989). Catastrophic interference in connectionist networks: The sequential learning problem. In G. H. Bower (Ed.), Psychology of Learning and Motivation (Vol. 24, pp. 109–165).

[2] Ratcliff, R. (1990). Connectionist models of recognition memory: Constraints imposed by learning and forgetting functions. Psychological Review, 97(2), 285–308.

---

> ### Author Rebuttal · Authors · 2025-07-31
>
> We thank the reviewer for their insightful feedback. We are grateful that they found our research questions intriguing, our results surprising and novel, and our setup clearly described. They demanded more support for our claim about the dynamic nature of memorization (sequences constantly being forgotten and re-memorized), which we provide via new experiments. They also sought clarification on other claims in the paper and highlighted missing citations. We respond to their comments below.
>
> **[W1] Differences from real-world model pretraining.** The reviewer highlighted that our multi-epoch training setup is different from the training of much larger real-world models that are typically trained for much fewer epochs.
>
> As for model size, we found our results to be applicable up to 1B parameter scale. (details in response to Reviewer mUn5; omitted here due to length limits) We will include them in the final version.
>
> Moreover, in this work, we set a particular scope to understand memorization in a more synthetic setup. It would be quite interesting to extend it for real-world pretraining. For such large datasets, we need to measure semantic (as opposed to verbatim memorization), as information can recur in a paraphrased manner. We see our work as the first step to understanding these dynamics.
>
> Furthermore, real-world data may contain duplicates. [1] found instances of a 61-word sentence in C4 dataset, which was repeated over 60,000 times. [2] raised concerns about duplicate data in several social media datasets. Training on a dataset with such duplicates is similar to a multi-epoch training regime. While deduplication is done on large datasets, it is extremely computationally expensive, especially for detecting near-duplicates. Our setup applies to these duplicate sequences.
>
> [1] Lee et al., "Deduplicating Training Data Makes Language Models Better.", ACL 2022.
>
> [2] Mu et al., "Enhancing Data Quality through Simple De-duplication: Navigating Responsible Computational Social Science Research.", EMNLP 2024.
>
> **[W2; Q1] More support for the dynamic nature of memorization.** We showed that sequences continue to be forgotten and re-memorized throughout training (i.e., memorization is dynamic). The reviewer questioned the use of exact match and suggested using softer measures of forgetting. Following their suggestion, we track the fraction of times a sequence is sampled (out of 5 generations) with a temperature of 0.5. We call this sampling accuracy ($s_{acc}$). We also track the status of each sequence at a given timestep: 0 (forgotten), 1 (memorized), as found via exact match. Below we report our findings on the following three models (similar to Table 1): 0.5M (trained on 100 examples), 1M (200 examples), and 2M (400 examples).
> 1. We average status and $s_{acc}$ across sequences and plot them across epochs. The two plots show a trend similar to Fig. 1 (rising from 0 to 1) and almost overlap. This suggests that the assignment of sequence status using exact match agrees with this softer sampling accuracy metric.
> 2. The average $s_{acc}$ across the sequences with status 0 (forgotten) was about 10-15%, while the average $s_{acc}$ across the sequences with status 1 (memorized) was about 99%. This strongly suggests that sequences that tend to be forgotten, as detected by exact match, also tend to have a low chance of being generated by the model.
>
> We also mention Fig. 14 (Appendix), where we plot metrics for forgotten sequences. The BLEU scores have a median of about 0.7. This suggests that multiple n-grams mismatch (as opposed to just a few tokens) when a sequence is deemed forgotten by exact match.
>
> Also, we use exact match since softer measures are not relevant in the context of PII or copyright infringement, which are real-world memorization issues.
>
> We thank the reviewer for their comment, as our claim is even stronger with these additional insights. We will include these results in the final version.
>
> **[W3; Q3] Definitions of ‘pattern’ and ‘generalization’.** We clarify these definitions below.
>
> *Pattern.* Assume an ideal (oracle) machinery $M$ that can learn any data generation process most efficiently. Then, for a dataset with patterns, $M$ learns a model better than unigram statistics. By ‘pattern’, we refer to any structure in the dataset that can be exploited by $M$.
>
> The reviewer also expresses doubt on whether a repeating trigram like “In the year…” is a pattern. Since the trigram appears multiple times across the dataset, this shared context can be used to memorize sequences (e.g., making the prediction of “year” more likely given “in the”). Therefore, $M$ will utilize it, and hence it is indeed a pattern.
>
> This also addresses reviewer’s concern about interpreting Table 3 results. When WikiText is shuffled, any shared n-gram context across sequences is lost. A model that could memorize WikiText is unable to memorize this shuffled dataset (without patterns, the dataset is harder to memorize and requires a model with more capacity).
>
> *Generalization.* In the context of our paper, a model is said to have generalized when it is shown to generate correct completions for unseen sequences that share patterns with the training dataset. We detail this in section 4.3 and in response to W5 below.
>
> We will make these definition-related clarifications in the paper and include this discussion on the interpretation of results in the Appendix.
>
> **[W4; Q2] Clarification on “overthinking”.** The reviewer showed concern regarding the framing of our “overthinking” result, where a larger model takes longer to memorize certain datasets with weak or absent patterns.
>
> To provide more context for this result, we refer to section 4.3, where we show generation statistics for models trained on “Monotonic Increasing with Jumps” dataset. This dataset contains monotonically increasing sequences of single digits with a 15% chance of a random jump to any digit. We observe that these models (4M and 8M) have 97.3% and 98.1% probability of generating a monotonically increasing digit, larger than the 85% probability of that happening in the dataset, suggesting that the models tend to strongly follow an identified pattern, with the larger model having more affinity. While this difference in probabilities for 4M and 8M models is small, so is the difference in epochs (~80) it takes them to memorize this “with jumps” dataset (Fig. 5).
>
> Given these observations, a likely and simple explanation (Occam's razor) is that the delay in section 4.4 is caused by the pattern-seeking behavior, which is more prevalent in larger models. We do agree that this is a hypothesis, and we will soften the framing accordingly.
>
> We use the term “overthinking” not to anthropomorphize the LM, but rather to provide a relatable term that conveys the phenomenon.
>
> **[W5] Clarification on the narrative.** The reviewer raised doubts about the story described in the paper. Below, we reiterate the results and how they fit the overall narrative.
>
> Through this work, we highlight that memorization in LMs is pattern-seeking and call attention to its different aspects. To understand what governs memorization, we first check if it is a constant accumulation of sequences without forgetting, or if it is biased by recency. We find neither of these to be the case. (4.1) Next, via training models on real-world and synthetic datasets, we observe that models memorize different datasets at different rates, leading to dataset-specific memorization profiles. These datasets differ in the extent of patterns present in them, which is correlated with the memorization rate. (4.2) To understand whether the model *really* extracts patterns while memorizing the data, we test it on an unseen validation set that also shares the same patterns. We find that models trained on datasets with static lookups generalize, but the models trained on datasets with dynamic lookups do not. This suggests that pattern extraction happens for memorization in the former, while not in the latter. We also observe that models tend to apply patterns more often than they are present in the training dataset, suggesting a strong affinity, with larger models showing more of it. (4.3) Finally, we observe that larger models take longer to memorize datasets that have weak or no patterns. We suspect this is attributed to the pattern-seeking behavior. (4.4)
>
> In this way, we view that the results presented in the paper highlight different dimensions of our central thesis (pattern-seeking behavior in memorization).
>
> The reviewer also asked about a particular section:
> > In particular, I think the section on generalization to static v/s dynamic patterns does not fit well into a paper that is interested in understanding memorization.
>
> As mentioned above, we use generalization to test *when* models exploit patterns to memorize. If they generalize, they must have extracted patterns; otherwise, individual sequences were independently memorized without regard to shared structure. Therefore, testing generalization allows us to differentiate between such rote memorization and the extraction of deeper patterns (data generation process), which is in line with our central thesis.
>
> **[W6] Missing citations.** We thank the reviewer for suggesting additional related work. We provide the updated related work section in response to Reviewer eVay. (omitted here due to length limits). We will include it in the final version.
>
> > **[Q4]** Out of all the results that are presented in the paper, which one would you say is the central contribution of your paper, and the one that summarizes the paper the best?
>
> As mentioned for W5, we view our paper as highlighting different aspects of pattern-seeking behavior in memorization studied under a multi-epoch training regime. All results collectively convey that thesis. Since this core insight is founded in all results equally, no single result, on its own, fully summarizes the overall contribution.

---

> > ### Comment · Reviewer_eqVj · 2025-08-01
> > **Thank you for the rebuttal**
> >
> > Thank you to the authors for their rebuttal.
> > - I am glad to see an expansive list of related work. This addresses that concern.
> > - Thank you for clarifying the definitions of pattern and generalization. These definitions certainly make the paper better defined. Still, as I mentioned in my review, I think that with these definitions, the results are not entirely surprising (for example, that shuffling the words makes it harder to memorize/fit the sequences).
> > - The clarification of the narrative is welcome, and I would encourage the authors to also apply this to streamline their paper further.
> > - I acknowledge that the authors do not wish to anthropomorphize the model. Yet, some readers and especially future work might mistakenly do so upon reading this paper, so I suggest that a different word than "seeking" be used when describing the delayed onset of learning, if possible. However, this is just a minor technical point.
> > - The expansion of results to models up to 1B also greatly strengthens the paper, as do the results with soft metrics. I will still contend that using 100-1000 epochs is rather excessive and may skew takeaways.
> >
> > Overall, I find that the rebuttal is reasonably convincing. I will therefore raise my score to "3: borderline reject" keeping both the improvements and the lingering concerns in mind.

---

> > > ### Author Response · Authors · 2025-08-02
> > > **Thank you for the acknowledgement of our rebuttal**
> > >
> > > We thank the reviewer for acknowledging our rebuttal. We are glad that they find most concerns have largely been addressed. We appreciate all the feedback on the presentation and additional experiments. Incorporating it has strengthened our paper. We again thank the reviewer for their time, and we are happy to answer any further questions.

---

### Official Review · Reviewer_mUn5 · 2025-06-28

**Clarity:** 3
**Significance:** 4
**Originality:** 3
**Rating:** 5
**Confidence:** 4

**Summary:**

This paper investigates the dynamics of memorization in transformer-based language models, focusing on how memorization evolves during multi-epoch training. The authors challenge the view that memorization is either a monotonic accumulation of data or driven solely by recency. Through controlled experiments on WikiText and synthetic datasets, they show that memorization occurs in bursts and is closely tied to the pattern complexity of the data. Models tend to memorize data with simpler or more salient patterns earlier and may “overthink” datasets lacking clear structure—delaying memorization. The paper introduces the notion of pattern-dependent memorization and highlights an intriguing phenomenon: larger models tend to spend more time trying to extract structure before resorting to rote memorization.

**Questions:**

I list my concerns and questions in the "weaknesses" section.

**Ethical Concerns:**

["NO or VERY MINOR ethics concerns only"]

**Final Justification:**

Thanks for the detailed responses. All my concerns are addressed.

**Limitations:**

yes

**Paper Formatting Concerns:**

None.

**Quality:**

3

**Strengths And Weaknesses:**

Strengths:
- The paper offers a dynamic perspective on memorization by tracking how models evolve during training, rather than relying solely on final snapshots.

- The idea that memorization is pattern-seeking (not just size- or exposure-driven) is well-supported through a variety of datasets with controlled structural complexity.

- The observed "overthinking" behavior in larger models adds a novel insight into model capacity and inductive bias.

Weaknesses:
- The experimental setup focuses on relatively small models (mostly up to 12M non-embedding parameters), which may limit the direct applicability of findings to larger-scale LLMs.

- The definition of "pattern complexity" remains somewhat qualitative; a more quantitative or formal measure would make the analysis more generalizable.

---

> ### Author Rebuttal · Authors · 2025-07-31
>
> We thank the reviewer for their feedback. We are glad that they liked our experimental design and think that our claims are well-supported. Furthermore, we appreciate that they recognize our observations as novel insights into models’ inductive biases. They expressed a concern regarding the use of relatively small models and the definition of pattern complexity. Below, we respond to their queries by reporting results from new experiments and providing clarifications.
>
> **[W1] Model sizes.**
> > The experimental setup focuses on relatively small models (mostly up to 12M non-embedding parameters), which may limit the direct applicability of findings to larger-scale LLMs.
>
> We experiment with larger models to show evidence for the applicability of our findings to larger model scales.
>
> We replicated the setup in the paper and performed new experiments on model sizes 100M, 200M, 500M, and 1B on datasets “Every 7th token same (10% sequences)” and “Monotonic Increasing with Jumps”. We consistently observe that larger models take longer (aka overthinking) before starting to memorize sequences in these datasets (same as Fig. 5). Please note that these training runs demonstrating overthinking can be very computationally expensive for larger models, as they take much longer to start memorizing those datasets, limiting the model sizes we can include in the study. However, these results up to 1B scale conclude that overthinking does hold at large scales.
>
> We further trained 200M and 500M models on the following datasets (one at a time): “Monotonic Increasing”, “Monotonic Increasing with Jumps”, “In-Context Mapping”, and “Random Digits”. We observed different memorization profiles (same as Fig. 2), with “Random Digits” sequences being memorized last.
>
> Hence, differentiation and overthinking are both applicable at larger model scales. We will include these results in the final version of the paper.
>
> **[W2] Definition of ‘pattern complexity’.**
> > The definition of "pattern complexity" remains somewhat qualitative; a more quantitative or formal measure would make the analysis more generalizable.
>
> We thank the reviewer for making this important point. We would like to clarify that there are two different perspectives on pattern complexity:
> 1. Complexity of the underlying data-generation process / algorithm
> 2. Learnability / Memorizability of the pattern by the model using gradient-based methods
>
> Under the first perspective, for datasets with randomness (e.g., jumps, random repeats, etc.), the amount of randomness gives a gradation of complexity. For deterministic datasets (Monotonic Increasing, In-Context Mapping, etc.), the number of lines of the shortest data-generating program serves as a measure of complexity (Kolmogorov complexity). However, it is difficult to compare patterns across these two dataset classes.
>
> In our paper, we focus on the learnability perspective to pattern complexity, which is tougher to define independently of the model. Our work highlights that the model has different affinities for memorizing different patterns, thereby suggesting that different patterns have different learnability. We believe that our work will motivate further analysis into this aspect of pattern complexity. We will include this discussion in the final version.

---

> > ### Comment · Reviewer_mUn5 · 2025-08-05
> >
> > Thank you for the responses. While I appreciate the clarifications, I will maintain my original score for now.

---

> > > ### Author Response · Authors · 2025-08-05
> > > **Thank you for the acknowledgement of our rebuttal**
> > >
> > > We thank the reviewer for acknowledging our rebuttal. We are glad they found the clarifications helpful and appreciate their continued positive assessment of our work.

---

### Official Review · Reviewer_p7BT · 2025-07-01

**Clarity:** 3
**Significance:** 3
**Originality:** 3
**Rating:** 5
**Confidence:** 4

**Summary:**

This paper empirically studies “discoverable” memorization (given a prompt, can the model exactly reproduce the subsequent tokens as in the training set) in language models via training dynamics. The main claim is that the model ‘seeks patterns’ during training for memorization, and does so in stages, each stage corresponding to patterns of certain complexity. Further, memorization of a sequence is not dependent on its recency in the training process. Overall, memorization is not simply copying verbatim sequences, but based on a notion of pattern recognition. They use toy examples as well as real-world language data to validate their hypotheses on models of different sizes.

**Questions:**

1. Model hyperparameters: I am not sure I completely follow the logic in selecting model sizes: why is E/L ratio a good metric for choosing language models for memorization? In fact, why the emphasis on embedding parameters - is there reason to believe that these parameters directly affect memorization or generalization in language models? The reasoning in the paper seems to be that some open source models use E/L in some (arguably large) range, which I did not completely follow in the above context. Also, why keep the number of attention heads = 1? Overall, I encourage the authors to rewrite this section for more clarity on these choices.


2. "Result 1: Memorization is neither a constant accumulation of sequences nor recency-biased." - I understand that constant accumulation / recency bias might intuitively make sense, but not sure if this has been claimed earlier? Mainly because I am not sure how a neural net like Transformer would implement ‘constant accumulation’ of training sequences. Could the authors clarify if this indeed happens (or if this could never happen) in some setup?


3. The toy datasets used in the paper are interesting and do model many aspects of memorization: however I believe a little more in-depth study would help understand memorization dynamics more clearly. Concretely, take the example of monotonic increase with jumps: instead of just using 15% probability, maybe vary this probability from 10-50% and then see how dynamics are affected. Monotonic increase with jump and letter: maybe use 2/3/… letters instead of just 1 after each jump. Repeat 5 tokens ago: check for 4/6/8... tokens ago instead, etc.

I am also curious about shuffled-WikiText: is there a way to control the degree of ‘shuffled-ness’? Maybe “more locally / more globally” shuffled than on a sentence / sequence level (e.g. multi-sentence shuffling)? It would be interesting to see what the results are on varying this characteristic.

To be clear, I do not expect the authors to implement all of this, and these are just some suggestions on how the study could be improved and made more systematic to derive better conclusions.

4. Line 193: “We also observe two distinct type of curves: sudden jump vs gradual rise, which also characterizes the data the model is trained on” - could the authors clarify how it characterizes training data beyond being one of the 2 types of curves?

**Ethical Concerns:**

["NO or VERY MINOR ethics concerns only"]

**Final Justification:**

My review was mainly about some clarifying questions (on model hyperparameters etc.), and some suggestions for ablations to improve their experimental analyses. They have addressed both these aspects in their rebuttal, and hence I've raised my score to 5.

**Limitations:**

Yes

**Quality:**

3

**Strengths And Weaknesses:**

Strengths:

- Studies a timely and important problem of memorization in language models.
- Experiments are well-motivated and clearly formulated, using various types of toy datasets.
- Paper is well written and easy to follow - the result takeaways at the beginning of each section are useful.

Weaknesses:

- Figure 2 is difficult to read - could the authors improve the visibility of the text in the left portion of this figure?
- Some model hyperparameter choices are not clearly motivated in the paper, specifically the emphasis on embedding parameters in Sec 3 and using only 1 attention head per layer.
- While the toy datasets used are interesting and relevant to memorization, I believe these could be studied a little more systematically. Please see Questions section below for discussion on this and other issues that are not exactly weaknesses.

---

> ### Author Rebuttal · Authors · 2025-07-30
>
> We thank the reviewer for their insightful feedback and suggestions to improve the quality of our work. We are glad that they found our experiments well-motivated and liked the presentation of the paper. The reviewer raised a concern regarding some hyperparameter choices and suggested further experimentation with our datasets. Below, we provide results from new experiments that further strengthen our claims. We will include these results in the final version of the paper.
>
> **[W1] Figure 2 visibility.** We shall update the figure to be more readable and also include the same text in the Appendix.
>
> **[W2; Q1.1] Clarification regarding embedding parameters and selecting model configurations.**
> > Some model hyperparameter choices are not clearly motivated in the paper, specifically the emphasis on embedding parameters in Sec 3…
>
> > I am not sure I completely follow the logic in selecting model sizes: why is E/L ratio a good metric for choosing language models for memorization?
>
> We believe there is a misunderstanding of what $E$ denotes. It is the size of the hidden dimension, i.e., model width. We will clarify this in the paper. We select model configurations for different parameter counts by maintaining a width-over-depth ratio ($E/L$) in the same range as present-day models like Llama 3.
>
> **[W2; Q1.2] Number of attention heads.**
> > Also, why keep the number of attention heads = 1?
>
> In our initial experiments, we observed that the number of attention heads does not impact the results, and hence we chose to use a single attention head. We did more experiments with models having 12 attention heads and found that our results still hold. Specifically, we train a 4M-parameter model having 12 attention heads on 500 examples each from “Monotonic Increasing” and “Monotonic Increasing with Jumps” datasets. We observed different memorization profiles (same as Fig. 2). We also trained models with 4M and 8M parameters on 500 examples of “Random Digits” data and noticed delayed memorization in the 8M model (like Fig. 5). Hence the main results of our study (differentiation and overthinking) are not impacted by a change in the number of attention heads. We will include these results and clarify the choice of the number of heads in the paper.
>
> **[Q2] Clarification on constant accumulation.** The reviewer asks if we observe constant accumulation of training sequences in some setup and seeks further clarification on this aspect. Firstly, we would like to reiterate that by constant accumulation, we mean a non-dynamic memorization where a sequence once memorized is never forgotten. This hypothesis is implicit in how the memorization of 'Personally Identifiable Information’ (PII) is often framed in discussions around training on such sequences, since the model potentially processes such information only once in the large pretraining corpus. We show that even in the setting most friendly to constant accumulation (training on the same data for multiple epochs), it is not observed in our setup. The other prevailing hypothesis could be that memorization is governed by the recency of sequences seen, which we also show to not be the case.
>
> **[W3; Q3.1] Further studies using toy datasets.** We thank the reviewer for suggesting experiments with different variations of our toy datasets. We present the results from these experiments with 4M models trained on 500 examples from the indicated datasets.
>
> > Monotonic Increasing with Jumps: vary the jump probability from 10-50% and then see how dynamics are affected
>
> We check for the following jump probabilities (in %): 10, 15, 20, 40, 50. As the probability of jumps increases (thereby increasing randomness), it takes longer to memorize the dataset, which is consistent with our claims.
>
> > Monotonic Increasing with Jumps and Letters: use 2/3/… letters instead of just 1 after each jump
>
> When more letters are used after the jump, memorization is faster. As the sequences are of finite length, with an increased number of monotonically increasing letters, the randomness decreases, making the sequences easier to memorize.
>
> > Repeat 5 tokens ago: check for 4/6/8... tokens ago instead
>
> Exact Match curves for all these runs (4/5/6/8 tokens ago) jump from 0 to 1 in the range of [230, 280] epochs without any particular order. This denotes that there is no significant difference in the complexity of these datasets in terms of memorizability.
>
> These results further strengthen our observations, and we will include these additional studies in the appendix.
>
> **[W3; Q3.2] Controlling the degree of ‘shuffled-ness’.**
> In the paper (Table 3), we show that the memorization of the dataset drops by 30% when the WikiText data is shuffled at the word level. The reviewer suggested varying the degree of shuffled-ness between the original and word-level shuffled states. As suggested, we shuffle at the sentence and multi-sentence levels. Moreover, we try a shuffle variant where we preserve the 2- and 3-grams in the top 50th percentile (in terms of occurrence counts) as one unit and shuffle them with the rest of the sequence. This lets us control the extent of patterns in the dataset. We present the ‘exact match’ results on 0.5M (trained on 100 examples) and 1M (trained on 200 examples) models. Consistent with our claims, we observe that the memorization capacity of a model is dependent on the extent of exploitable structure present in the data. With reduced structure (increased degree of shuffled-ness), the model memorizes less data. We will extend Table 3 with these results.
>
> | Shuffle variant                                      | 100 ex; 0.5M | 200 ex; 1M |
> |-------------------------------------------------|-------------------|-----------------|
> | Baseline (original WikiText)                                        | 0.96              | 0.96            |
> | Shuffle 2-sentence units                      | 0.94              | 0.95            |
> | Shuffle sentences                        | 0.93              | 0.92            |
> | Preserve top 50% 3-grams          | 0.78              | 0.78            |
> | Preserve top 50% 2-grams          | 0.72              | 0.73            |
> | Word-level shuffled                                        | 0.68       | 0.63            |
>
>
>
> > **[Q4]** Line 193: “We also observe two distinct type of curves: sudden jump vs gradual rise, which also characterizes the data the model is trained on” - could the authors clarify how it characterizes training data beyond being one of the 2 types of curves?
>
> This is a typographical error on our part. We will correct it as follows: “We also observe two distinct type of curves: sudden jump vs gradual rise, which is another characteristic feature of the data the model is trained on”.

---

> > ### Comment · Reviewer_p7BT · 2025-08-01
> > **Thanks for the rebuttal**
> >
> > The authors' response has clarified my questions, and their new results have improved the submission. I've raised my score to 5.

---

> > > ### Author Response · Authors · 2025-08-02
> > > **Thank you for the acknowledgement of our rebuttal**
> > >
> > > We thank the reviewer for acknowledging our rebuttal and for increasing the rating. We are glad that our response clarified their concerns and that they believe the submission has improved, given new results.

---

### Official Review · Reviewer_eVay · 2025-07-02

**Clarity:** 2
**Significance:** 1
**Originality:** 2
**Rating:** 4
**Confidence:** 3

**Summary:**

This paper investigates the dynamic and pattern-driven mechanisms of memorization in Transformer-based language models. The authors demonstrate that memorization is not a simple accumulation of training sequences nor determined by recency of exposure. It is driven by pattern recognition.

**Questions:**

Are there any reasons for choosing GPT-2 (2019) for the experiments? Typically, smaller models are chosen for interpretability reasons, but I didn't see a thorough interpretability study in the paper.

**Ethical Concerns:**

["NO or VERY MINOR ethics concerns only"]

**Final Justification:**

I updated my score because the authors provide the detailed related work and address my concern about the model size

**Limitations:**

Yes

**Paper Formatting Concerns:**

No formatting issue.

**Quality:**

2

**Strengths And Weaknesses:**

Strengths:
- The authors validate that memorization in Transformer models is pattern-driven rather than simply a matter of sequence accumulation or recent exposure.
- The memorization behavior of the models is demonstrated to be driven by patterns rather than gradual accumulation or recency.
- The authors find the "overthinking" phenomenon, in which larger models exhibit delayed memorization on data.

Weaknesses:
- The largest model used in the experiments is only 85M parameters, this is hard to conclude that "overthinking" would also occur in larger models.
- The paper lacks recent literature. I suggest that the authors pay more attention to recent papers on the understanding of memorization in language models, such as: https://arxiv.org/abs/2404.15146, https://arxiv.org/abs/2411.07175, https://arxiv.org/pdf/2404.13501, and https://arxiv.org/abs/2504.12523 etc.

---

> ### Author Rebuttal · Authors · 2025-07-31
>
> We thank the reviewer for highlighting the strong results of our work, such as memorization being driven by patterns and larger models exhibiting delayed memorization (overthinking). They had a concern about the relatively small model sizes used in the paper and about the paper missing some references. We ran additional experiments, which further support our results and provide an updated section on related work. We will include them in the paper to make the resulting manuscript even better. We respond to their comments below.
>
> **[W1; Q1] Model sizes.** In our work, we find that larger models show delayed memorization on data with weak patterns. We call it overthinking. The reviewer expressed a concern regarding the validity of this result at larger model scales. We replicated the setup in the paper and performed new experiments on model sizes 100M, 200M, 500M, and 1B on datasets “Every 7th token same (10% sequences)” and “Monotonic Increasing with Jumps”. We consistently observe that larger models take longer before starting to memorize sequences in these datasets (same as Fig. 5). We note that these training runs demonstrating overthinking can be very computationally expensive for larger models, as they take much longer to start memorizing those datasets, limiting the model sizes we can include in the study. However, these results up to 1B scale conclude that overthinking does hold at large scales. We will include them in the final version.
>
> **[W2] Missing references.** The reviewer highlighted certain related work that is missing from the paper. We thank the reviewer for bringing these references to our attention. Below, we provide the updated related work section that covers all suggested references. We shall include it in the final version, making the related work coverage more holistic.
>
> ## Related Work
> Memorization has been a central topic in machine learning since the earliest neural networks. We organize prior work on this topic into six themes: (1) classical and associative memory models, (2) memorization in modern language models, (3) privacy and data extraction, (4) memorization versus generalization and pattern-seeking behavior, (5) interpretability and training dynamics, and (6) related learning phenomena such as forgetting and curricula.
>
> ### Classical and Associative Memory Models
> The study of neural associative memories began with Hopfield networks [1], which showed that recurrent networks could store discrete patterns as stable attractors. These systems, along with bidirectional associative memories [2] and sparse distributed memory [3], were designed to explicitly encode and retrieve stored patterns from the network’s weights, leading to a conception of neural networks as content-addressable memories.
>
> While modern transformers use attention and contextual representations rather than discrete attractors, these classical models highlight that the ability to store and recall information has always been a fundamental property of neural architectures. Recent surveys emphasize that modern LLMs behave differently from these classical models, exhibiting implicit memory through distributed representations, attention mechanisms, and in-context learning rather than explicit associative storage [4]. This shift motivates a fresh look at memorization in the transformer era.
>
> ### Memorization in Transformer Language Models
> In modern models, memorization is less deliberate but equally real. Large-scale studies demonstrate that transformers memorize exact sequences from their training data. Starting with the early extraction experiments in GPT-2 [5], research has repeatedly shown that certain examples (especially those that are unique or duplicated) can be reproduced verbatim during inference [6, 7, 8, 9]. Empirical evidence indicates that memorization grows with model scale, dataset duplication, and training length. Strategies such as deduplicating training data [10] reduce this tendency.
>
> Recent work proposes alternative ways of thinking about memorization, including counterfactual memorization [11] (asking what a model would output if a particular example were removed from training) and adversarial compression views [12], which link memorization to how models compress information under overparameterization. Although our focus is on language models, memorization has been observed across modalities: for instance, extraction from diffusion models [13], code models [14], and large vision models [15] demonstrates that this is a general property of overparameterized models.
>
> ### Privacy and Data Extraction
> Memorization directly affects privacy and intellectual property. Data extraction studies [5, 16, 6, 7, 9] have shown that sensitive information such as names, addresses, or copyrighted material can be recovered from trained models, even without direct access to their training data. Large-scale attacks on production LLMs [7] have scaled these demonstrations, showing that thousands of training examples can be extracted at low cost, and work on model stealing [17] has shown that the problem persists even when the adversary only queries a deployed model. These results motivate defenses: deduplication [10] and differential privacy-based training can reduce leakage, though at the expense of performance. The risk of leakage is not limited to NLP models but also applies to other modalities [13, 14, 15].
>
> ### Memorization vs. Generalization and Pattern-Seeking Behavior
> While memorization is often cast as the opposite of generalization, work from deep learning has shown that the relationship is more nuanced. Networks can memorize random labels [18], but in structured datasets, models first capture simpler patterns and only later memorize irregularities [19]. This pattern-seeking inductive bias is particularly important in LMs: when patterns exist, models generalize; when they do not, models ultimately memorize specific examples. Evidence for such biases includes frequency-driven acquisition of syntactic rules [20], mapping of a memorization$-$generalization continuum [21], and studies on the long tail, where rare examples are memorized rather than generalized [22].
>
> Recent taxonomies [23] emphasize that memorization itself has multiple modes, from simple recitation of exact strings to reconstruction based on partial patterns, suggesting that memorization and generalization can co-exist within the same model. Complementary perspectives view memorization as a form of compression [12], and phenomena like grokking [24] illustrate that generalization can appear abruptly after long periods of apparent overfitting, driven by latent structure-seeking. Our results echo these insights: memorization does not progress monotonically but is shaped by perceived pattern difficulty.
>
> ### Interpretability and Training Dynamics
> Another line of work studies when and how memorization occurs. Training-dynamics analyses have found phase transitions in LMs: loss drops accompanied by sudden acquisition of syntactic or structural knowledge [25]. Representation studies [26] have shown that language models acquire features (syntax, semantics) in a progressive, structured order, which may explain the order in which they memorize or generalize. Beyond training, post-hoc interventions attempt to control knowledge in already-trained models; recent work examines how to update models with new knowledge and balance memorization with reasoning [27]. These threads converge on the idea that memorization is not just an endpoint but a process: understanding it requires looking at how representations and behaviors evolve during training.
>
> ### Related Learning Phenomena: Forgetting and Curricula
> Memorization is closely connected to forgetting and curricula. Catastrophic forgetting, where models overwrite previously learned information, was first documented in early connectionist systems [28, 29]. It remains an issue even in large models; fine-tuning can cause previously memorized examples to be forgotten [8]. New approaches, such as continual memorization [30], highlight the need to balance stability and plasticity.
>
> Data ordering also shapes memorization: curriculum learning [31] biases models to learn easier, pattern-rich examples first, postponing rote memorization. These observations resonate with our findings that transformer LMs, even without explicit curricula, exhibit a natural ordering in what they memorize, preferring structured patterns first.
>
> **Following are all the references as DOIs. For arXiv papers, the DOI is the Crossref form 10.48550/arXiv.XXXX.XXXXX. For books / journal articles, the original DOI is used where available.**
>
>
> [1] 10.1073/pnas.79.8.2554
>
> [2] 10.1109/21.87054
>
> [3] Kanerva, P. (1988). Sparse Distributed Memory. MIT Press.
>
> [4] arXiv.2404.13501
>
> [5] arXiv.2012.07805
>
> [6] arXiv.2202.07646
>
> [7] arXiv.2311.17035
>
> [8] arXiv.2207.00099
>
> [9] arXiv.2210.17546
>
> [10] arXiv.2107.06499
>
> [11] arXiv.2112.12938
>
> [12] arXiv.2404.15146
>
> [13] arXiv.2301.13188
>
> [14] arXiv.2308.09932
>
> [15] arXiv.2310.05337
>
> [16] arXiv.2206.10469
>
> [17] arXiv.2403.06634
>
> [18] arXiv.1611.03530
>
> [19] arXiv.1706.05394
>
> [20] arXiv.2109.07020
>
> [21] arXiv.2311.05379
>
> [22] arXiv.2008.03703
>
> [23] arXiv.2406.17746
>
> [24] arXiv.2201.02177
>
> [25] arXiv.2309.07311
>
> [26] arXiv.1811.00225
>
> [27] arXiv.2504.12523
>
> [28] 10.1016/S0079-7421(08)60536-8
>
> [29] 10.1037/0033-295X.97.2.285
>
> [30] arXiv.2411.07175
>
> [31] 10.1145/1553374.1553380

---

> > ### Comment · Reviewer_eVay · 2025-08-05
> >
> > The authors address most of my concerns and I will raise my score to borderline accept

---

> > > ### Author Response · Authors · 2025-08-05
> > > **Thank you for the acknowledgement of our rebuttal**
> > >
> > > We sincerely thank the reviewer for acknowledging our rebuttal and updating their score to a 4. We are pleased that most of their concerns have been addressed and would be happy to clarify any remaining questions that might further inform their recommendation.

---

### Decision · Program_Chairs · 2025-09-17

**Decision:**

Accept (poster)

**Comment:**

This paper investigates how transformer-based language models memorize information. The authors argue that memorization isn't just about storing data but is a process driven by pattern recognition, similar to how models generalize. Through experiments, they show that models learn in bursts, tackling simpler patterns first before moving to more complex ones. A key finding is a phenomenon they call "overthinking," where larger models delay memorizing random or pattern-less data, seemingly because they are searching for underlying structures to learn instead.

Initially, the reviewers had several key concerns. These included the use of small models, a lack of both recent and foundational related work, and the use of vague terms such as "pattern." They also questioned the experimental setup, such as training for an excessive number of epochs and using a strict metric for forgetting. In response, the authors expanded their experiments to include models up to 1B parameters, added a comprehensive literature review, and provided clearer definitions. These changes addressed the majority of concerns, leading three of the four reviewers to raise their scores to "Accept" or "Borderline Accept." One reviewer, while acknowledging the improvements and some resolved issues, still have remaining concerns about the training setup and that terminologies used. I think the authors should be able to find what the best terminology is to improve their presentation to the readers.